# Competence for transcellular infection in the root cortex involves a post-replicative, cell-cycle exit decision in *Medicago truncatula*

Morgane Batzenschlager[1]*, Beatrice Lace[1], Ning Zhang[1,2†], Chao Su[1‡], Anna Boiger[1], Sabrina Egli[1,3§], Pascal Krohn[1#], Jule Salfeld[1], Franck Anicet Ditengou[1¶], Thomas Laux[1,2], Thomas Ott[1,2]*

[1]Faculty of Biology, University of Freiburg, Freiburg im Breisgau, Germany; [2]CIBSS – Centre of Integrative Biological Signalling Studies, University of Freiburg, Freiburg im Breisgau, Germany; [3]CEPLAS – Cluster of Excellence on Plant Sciences, Heinrich-Heine-University Düsseldorf, Düsseldorf, Germany

**\*For correspondence:** morgane.batzenschlager@ biologie.uni-freiburg.de (MB); Thomas.Ott@biologie.uni-freiburg.de (TO)

**Present address:** †State Key Laboratory of Wheat Improvement, College of Agronomy, Shandong Agricultural University, Tai'an, China; ‡National Key Laboratory of Crop Genetic Improvement, College of Plant Science and Technology, Huazhong Agricultural University, Wuhan, China; §Institute of Cell and Interaction Biology, Heinrich-Heine-University Düsseldorf, Düsseldorf, Germany; #Department for Plant-Microbe Interactions, Max Planck Institute for Plant Breeding Research, Cologne, Germany; ¶Institute for Disease Modelling and Targeted Medicine (IMITATE), Freiburg University Medical Centre, Freiburg, Germany

**Competing interest:** The authors declare that no competing interests exist.

**Preprint posted** 14 April 2023 **Sent for Review** 21 April 2023 **Reviewed preprint posted** 19 July 2023 **Reviewed preprint revised** 02 June 2025 **Version of Record published** 04 July 2025

## eLife Assessment

This is a **fundamental** cell biological study of host responses during symbiotic microbial infection of plants. **Compelling** imaging-based approaches using genetically encoded cell cycle markers show that in Medicago truncatula root cortex cells, early rhizobial infection events are associated with cell-cycle re-entry, but once the infection is established, host cells exit the cell cycle. The work will be of interest to a wide range of readers working in fields from development and cell biology to plant-microbe interactions.

**Abstract** During root nodule symbiosis (RNS), cell-division activity is reinitiated and sustained in the root cortex to create a hospitable cellular niche. Such a temporary and spatially confined site is required to render host cells compatible with the intracellular progression of rhizobia. Although it has been suggested that early infection events might involve a pre-mitotic cell-cycle arrest, this process has not been dissected with cellular resolution. Here, we show that a dual-color *Medicago* histone reporter robustly identifies cells with different mitotic or endoreduplication activities in the root cortex. By imaging deep root tissues, we found that a confined trajectory of cortical cells that are transcellularly passed by infection threads is in a stage of the cell cycle that is distinct from directly adjacent cells. Distinctive features of infected cells include nuclear widening and large-scale chromatin rearrangements consistent with a cell-cycle exit prior to differentiation. Using a combination of fluorescent reporters demarcating cell-cycle phase progression, we confirmed that a reduced proliferation potential and modulating the G2/M transition, a process possibly controlled by the NF-YA1 transcription factor, mark the success of rhizobial delivery to nodule cells.

## Introduction

Cell-cycle regulation is a key pathway to maintain beneficial and sustainable symbiotic associations that are of primary importance in both marine and terrestrial ecosystems. One common theme to stabilize facultative associations is to achieve a mutual control of cell proliferation and differentiation dynamics in both the host and the symbionts (*Kondorosi and Kondorosi, 2004*; *Russo and Genre, 2021*). Prime examples are flowering plants from a single phylogenetic clade, comprising four

orders (Fabales, Fagales, Cucurbitales, Rosales) with legumes being the most prominent ones, that evolved the unique ability to accommodate soil-borne rhizobia intracellularly (*Parniske, 2018*). Within this symbiosis, rhizobia fix atmospheric nitrogen and deliver ammonium to the host in exchange for plant photosynthates. The conversion of atmospheric dinitrogen into a usable form is highly energy-demanding and requires a low-oxygen environment, which is provided within nodules, specialized lateral organs formed on the roots that are densely populated by rhizobia. A widely adopted mechanism to access the developing nodule cells is the formation of cortical infection threads, transcellular conduits formed upon the invagination of the host plasma membrane and hijacking the cellular machinery of infected root cortical cells (*Timmers et al., 1999*; *Su et al., 2023*). Although the molecular and cellular understanding of the infection mechanisms is expanding (*Tsyganova et al., 2021*), our knowledge on how exactly these processes are interconnected with the control of cell-cycle progression in cells hosting intracellular infection threads has remained fragmentary.

Plants engaging in arbuscular mycorrhizal (AM) or root nodule symbioses (RNS) have recruited a common range of cell-cycle processes (*Foucher and Kondorosi, 2000*; *Russo and Genre, 2021*). In *Medicago truncatula* (*M.t.*; hereafter, Medicago), the perception of and colonization by the AM fungus result in occasional cell divisions in the inner cortical layers (*Russo et al., 2019*) and in diffuse endoreduplication events as intraradical hyphae spread throughout the root cortex (*Carotenuto et al., 2019a*; *Carotenuto et al., 2019b*). During RNS, the perception of rhizobia results in the onset of a signaling cascade that triggers an outward gradient of cell-cycle reactivation and cell divisions. Although this is initiated in the pericycle, cell-cycle reactivation occurs throughout the inner cortical (C) layers C5, C4, and the middle C3 layer in Medicago (*Timmers et al., 1999*; *Xiao et al., 2014*). The successful establishment of RNS further involves the tight coordination of the infection and organogenesis programmes (*Guan et al., 2013*). It requires the progression of cortical infection threads through the C3 layer prior to subsequent periclinal divisions that produce a persistent nodule meristem (*Xiao et al., 2014*). The resulting indeterminate nodules will later form well-defined developmental zones, with actively dividing meristematic cells giving rise to post-meristematic, differentiating central tissues of increasing ploidy (*Vinardell et al., 2003*; *Nagymihály et al., 2017*). While those cells can be intracellularly colonized, the infection competence of nodule-like structures as being induced by exogenous application of cytokinin (*Gauthier-Coles et al., 2019*), constitutive activation of common symbiotic components (*Gleason et al., 2006*; *Singh et al., 2014*) or the expression of meristematic factors (*Dong et al., 2021*) may be massively reduced in many of these spontaneous nodules (*Hayashi et al., 2010*; *Liu et al., 2022*). The lack of infectibility cannot be overcome even when applying bacterial strains which enter plant cells from the intercellular space (*Liang et al., 2019*). This clearly demonstrates the ultimate requirement of cells being in a so far uncharacterized state of infection competence.

Transcriptome profiling of Medicago root hairs, treated with rhizobial Nod factors or undergoing rhizobial colonization, uncovered changes in several cell-cycle-related genes (*Breakspear et al., 2014*; *Liu et al., 2019a*). This perception of Nod factors results in the increased transcription of D-type cyclins, genes known to respond to extrinsic signals and to promote cell-cycle progression from the first GAP (G1) phase to the DNA-synthesis (S) phase (*Dewitte et al., 2003*). Subsequently, the initiation of infection threads coincides with the highest expression of genes required for DNA replication, modification, and repair, strongly suggesting that the epidermal infection programme involves cell cycle re-entry and progression to a post-replicative phase (*Breakspear et al., 2014*). A repression of the endocycle was further proposed to occur since infected root hairs accumulate transcripts of the *OSD1/UVI4* gene (*Breakspear et al., 2014*), whose activity prevents an unscheduled increase in ploidy by inhibiting the anaphase-promoting complex/cyclosome (*Iwata et al., 2011*). However, much less is known about the transcriptional signature of infected cells along the cortical trajectory of transcellularly progressing infection threads. Such ability to guide rhizobia *via* cortical infection threads is yet thought to be an important innovation since it has been evolutionary maintained in the vast majority of nodulating species as it, most likely, maximizes the host control over bacterial delivery (*Parniske, 2018*; *Cathebras et al., 2022* – pre-print).

A central regulator for controlling infection and organogenesis is the transcription factor *Nodule Inception* (*NIN*). Co-opted from nitrate response circuits, NIN acts as a master coordinator of infection, nodule organogenesis, and nodule number (*Marsh et al., 2007*; *Fournier et al., 2015*; *Soyano et al., 2014*; *Cathebras et al., 2022* – pre-print). It promotes the expression of several growth targets (*Liu et al., 2019a*)

including *NF-YA1* (*Soyano et al., 2013*; *Feng et al., 2021*), a subunit of the conserved heterotrimeric Nuclear Factor-Y (NF-Y) transcription factor. A-, B-, and C-type subunits of the NF-Y complex act as specialized modules, interacting with other transcription factors and recruiting chromatin remodeling enzymes to adjust cell specification to environmental or developmental needs (*Zanetti et al., 2017*; *Myers and Holt, 2018*). In vertebrates, the CCAAT-binding NF-Y transcription factor targets genes involved in all major activities executed in G2 and mitosis (M; *Linhart et al., 2005*) and is an important regulator of the G2/M transition (*Manni et al., 2001*). In legumes, the rhizobium-induced NF-YA1 subunit functions in cortical infection thread progression, in the establishment and maintenance of the nodule meristem and in the early differentiation of nodule cells (*Combier et al., 2006*; *Laporte et al., 2014*; *Xiao et al., 2014*; *Hossain et al., 2016*; *Lee et al., 2024*). Yet, whether the Medicago NF-YA1 protein regulates the G2/M transition is not known.

The hypothesis that intracellular infection at least partially relies on cell-cycle control is further supported by structural rearrangements in cortical cells anticipating transcellular infection thread progression. These include the formation of pre-infection threads (PITs), which are transvacuolar, cytoplasmic bridges comparable to those occurring in pre-mitotic cells (*van Brussel et al., 1992*; *Timmers et al., 1999*). In situ detection of marker transcripts revealed that outer cortical cells recruit histone H4 used during DNA replication but do not express a B-type, mitotic cyclin gene – implying that activated target cells stop in the G2 phase of the cell cycle (*Yang et al., 1994*). G2-phase cell-cycle arrests have been repeatedly described in eukaryotic cells as appropriate gates for executing cell fate and patterning decisions (*Meserve and Duronio, 2017*), for wound healing (*Cosolo et al., 2019*) or for host infection during pathogenic and biotrophic interactions (*Wildermuth et al., 2017*). Modulating the G2/M transition combines well-described transcriptional and post-translational mechanisms, leading to the repression of genes necessary to execute cell division (*Berckmans and De Veylder, 2009*; *Kobayashi et al., 2015*), controlling the activity of cyclin-dependent kinases (CDK) and the stability of transcriptional repressors and mitotic inducers (*Kondorosi and Kondorosi, 2004*; *Chen et al., 2017*). Although previous reports have suggested such cell-cycle patterns to occur in roots during rhizobial infections (*Yang et al., 1994*; *Breakspear et al., 2014*), this has never been spatially resolved at the cellular level.

Here, we used a combination of cell-cycle reporters to resolve functional steps of cell-cycle progression on the cortical infection thread trajectory with cellular and subcellular resolution. We demonstrate that the first generation of Medicago cells successfully internalizing rhizobia shows unique features, including a marked decrease in their histone H3.1 content and a reduced competence for both cell division and chromosome segregation. We also show that tetraploid (4n) cells reaching an 8C DNA content are optimal for infection by rhizobia. Our results strongly suggest that, upon transcellular infection thread passage, cortical cells undergo a last GAP phase after DNA replication and exit to differentiation. Furthermore, we present the first evidence that the symbiosis-induced NF-YA1 subunit, by controlling mitotic entry in a heterologous system and ectopically in transgenic Medicago roots, holds potential to contribute to this process.

## Results

### A reduced proliferative potential typifies cells supporting cortical infection thread progression

In *Arabidopsis thaliana* (*A.t.*; hereafter, *Arabidopsis*), the incorporation and eviction dynamics of histone H3 variants have been used to identify cells with different cell division and reprogramming potentials, during organ patterning (*Otero et al., 2016*) or when acquiring reproductive competence (*Hernandez-Lagana and Autran, 2020*). To test whether gaining a stage of intracellular infection competence involves a similar modification of the cell-division potential and associated chromatin reorganization, we searched for Medicago genes coding for the replicative histone variant H3.1 and the replacement variant H3.3 (*Figure 1—figure supplement 1A*; *Probst et al., 2020*). The encoded Medicago H3.1 protein is identical to the one present in *Lotus japonicus* and *Hordeum vulgare* and, as in other flowering plants, differs from H3.3 by only four residues (*Figure 1—figure supplement 1B*; *Shi et al., 2011*). To check that these Medicago *H3* genes encode for *bona fide* H3.1 and H3.3 counterparts, native promoter-genomic fusions with fluorescent tags were expressed in *Agrobacterium rhizogenes*–induced transgenic roots, formed on composite plants that were inoculated with

the compatible symbiont *Sinorhizobium meliloti* (each composite plant representing an independent transformant; *Boisson-Dernier et al., 2001*). We adopted a fixation, clearing, and cell-wall counter-staining procedure (*Ursache et al., 2018*) that enabled us to perform detailed imaging deep inside the multi-layered root cortical tissues. Medicago H3 fusion proteins fully recapitulated the patterns observed in *Arabidopsis* roots (i.e. patchy pattern of H3.1 and constitutive presence of H3.3; *Figure 1—figure supplement 1C–E*) and showed the expected localization in euchromatin (diffuse labeling) and heterochromatic regions (subnuclear foci; *Figure 1C*, *Figure 1—figure supplement 1E*; *Ingouff et al., 2010*; *Shi et al., 2011*; *Otero et al., 2016*). A low H3.1 content, typical for cells with a reduced proliferative potential, also enabled us to identify putative quiescent centre (QC) cells within the open root apical meristem of Medicago (*Figure 1—figure supplement 1E*; *Rost, 2011*; *Otero et al., 2016*; *Xiao et al., 2019*). Preferentially incorporated during DNA replication, the canonical histone H3.1 variant controls the maintenance of genome integrity (*Davarinejad et al., 2022*) and epigenetic inheritance (*Jiang and Berger, 2017*) and is consequently maintained at high levels in cells engaged in recurrent cell division or endoreduplication cycles (*Figure 1A*). Once cells exit the cell cycle, H3.1 is evicted and likely replaced by H3.3 to mediate cellular differentiation in various *Arabidopsis* cell types (*Otero et al., 2016*). In Medicago, increased levels of *H3.1* transcripts were coherently detected at an early stage of nodule primordia development (*Figure 1—figure supplement 1F*; *Schiessl et al., 2019*) and mRNAs accumulated only in meristematic (ZI) and infection zones of mature nodules (ZIId and ZIIp; *Roux et al., 2014*), while *H3.3* transcription stayed high in the differentiation zone (ZIII; *Figure 1—figure supplement 1G*). We verified a corresponding differential accumulation of H3.1 and H3.3 tagged proteins at cellular resolution in nodule sections (*Figure 1B*). The double histone reporter proved also useful to distinguish actively endoreduplicating cells prior to fungal colonization (*Carotenuto et al., 2019a*; *Carotenuto et al., 2019b*) from fully differentiated, arbuscule-containing cells (*Figure 1C*). Since the selected Medicago histone variants recapitulated the expected transcriptional and localization patterns in well-defined developmental zones, we considered them as functional reporters of H3.1 and H3.3. In addition, the H3.1/H3.3 balance enables tracking cortical cells re-entering the cell cycle and adapting their responses to both bacterial and fungal symbionts.

We then focused on the H3.1 nuclear content at early interaction stages between rhizobia and the host plant (8 days post inoculation [dpi]; *Figure 2A–B*). In accordance with RNA sequencing data (*Figure 1—figure supplement 1F*; *Schiessl et al., 2019*), we observed the replicative H3.1 variant accumulating in regularly dividing cortical cells forming the nodule primordium (*Figure 2A*, upper panels). The H3.1-eGFP fusion protein was also visible in cells penetrated but not fully passed by an infection thread (*Figure 2A*, lower panels). This is consistent with earlier observations that cells preparing for intracellular infection pass through S-phase (*Yang et al., 1994*). By contrast, host cells sustaining cortical infection thread progression in the middle and inner cortex (C3 and C4) up to the first recipient cells in the nodule primordium (C4/5-derived; *Figure 2B*) exhibited enlarged nuclei (*Figure 2C*) with a significantly decreased histone H3.1 content compared to their direct neighboring cells (*Figure 2D*). Such a pattern was also observed using another *H3.1* gene expressed in root tissues (*H3.1 (2)*; *Figure 2—figure supplement 1A–E*). Nuclear enlargement associated with a massive eviction of H3.1 typically identifies cells having completed a DNA replication round, but pausing in the following GAP phase for cell-cycle switches or exit decisions (*Figure 1A*; *Otero et al., 2016*; *Hernandez-Lagana and Autran, 2020*; *Probst et al., 2020*). Together, our data strongly suggest that cortical cells along the infection thread trajectory, different from their direct neighbors, exhibit a reduced proliferation potential and presumably exit the cell cycle (*Figure 2B–D*). The observed nuclear enlargement and large-scale chromatin rearrangements are consistent with increased transcriptional activity (*Knaack et al., 2022*) concomitant with sustained infection thread progression (*Breakspear et al., 2014*; *Liu et al., 2019a*). This proves the long-standing hypothesis that cortical cells competent for transcellular infection suspend their course in G2 (*Yang et al., 1994*; *Timmers et al., 1999*) in Medicago.

## Exiting the cell cycle on the cortical infection thread trajectory aligns infection and organogenesis

Next, we tested whether a cell-cycle exit decision and associated remodeling of the histone H3 composition requires the tight coupling of infection and nodule organogenesis as observed under wild-type (WT) conditions. Thus, we compared the H3.1/H3.3 patterns in WT roots to those of two mutants

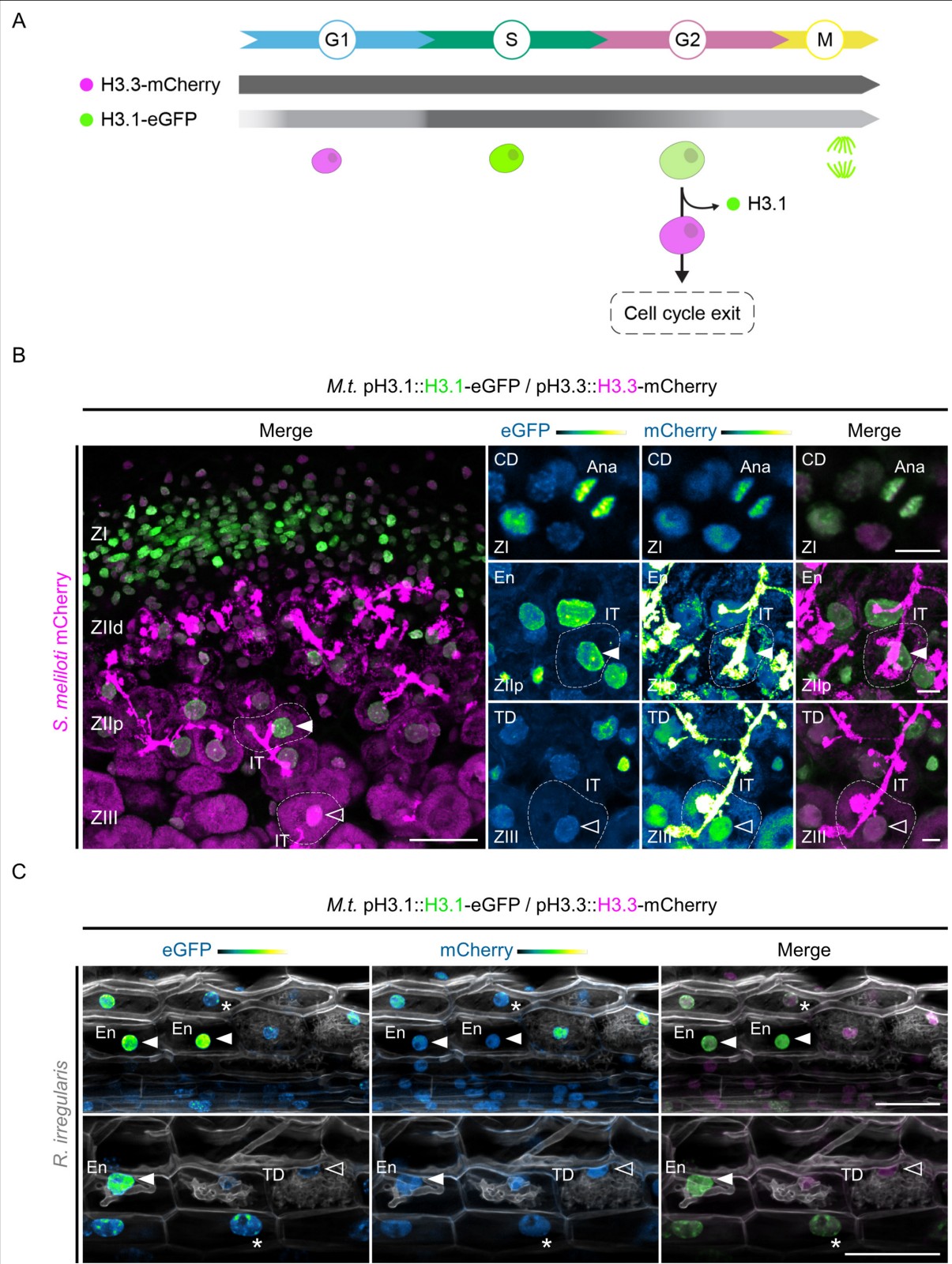

**Figure 1.** Medicago histone H3.1 patterns reveal sustained mitotic and endocycling activities in a symbiotic context. (**A**) Schematic representation of histone H3.3-mCherry and H3.1-eGFP distribution (horizontal bars) and fluorescence intensity (gray saturation) throughout the different cell-cycle phases (adapted from *Echevarría et al., 2021*). H3.1 is predominantly expressed during S-phase and incorporated during DNA replication in proliferating (G1, S, G2, M) and endocycling (G1, S, G2) cells. H3.3 is constitutively produced. Nuclei of increasing size and DNA content are colored according to

*Figure 1 continued on next page*

*Figure 1 continued*

their H3.3 (pink) and H3.1 (green) content in decondensed chromatin (G1, S, G2) or condensed chromosomes (M). This diagram also illustrates cells exiting the cell cycle within proliferating or endoreduplicating populations, which leads to significant H3.1 eviction following their last DNA replication round (*Otero et al., 2016*). (**B**) Confocal images of nodule sections isolated from WT transgenic roots inoculated with mCherry-producing *S. meliloti* (35–40 dpi), showing H3.1-eGFP (green) and H3.3-mCherry (magenta) localization across characteristic developmental zones. H3.1 accumulates in the meristematic zone (ZI) enriched in proliferating cells and in regions with high endoreduplication activity (ZIId and ZIIp, distal and proximal parts of the infection zone) where rhizobia are released into membrane-bound compartments called symbiosomes. H3.1 is extensively replaced by H3.3 in the fixation zone (ZIII) where host cells and rhizobia complete their differentiation process. IT: infection thread. Ana: late-anaphase chromosomes. Filled or empty arrowheads point to nuclei with a high (ZIIp) or a low (ZIII) H3-1-eGFP content, respectively. Dashed lines demarcate infected nodule cells containing symbiosomes. Images are maximum intensity projections except the top-right panels (single focal plane). Scale bars: left panel = 50 μm; right panels = 10 μm. Transformation experiments were repeated three times with a total of nine nodules from six composite plants showing similar results. (**C**) Confocal images of whole-mount transgenic roots colonized by *Rhizophagus irregularis* (15 dpi). Plant and fungal cell walls were stained with Calcofluor white (grayscale). H3.1 is enriched in chromocenters (heterochromatin foci in nuclei indicated by stars) and kept at high levels in the euchromatin (diffuse labeling in nuclei pointed by filled arrowheads) from neighboring (upper panel) and early-arbusculated cells (lower panel) of the inner cortical tissue. The empty arrowhead points to a nucleus with a low H3.1 content in a fully-arbusculated, differentiated cell. Scale bars: 50 μm. Two independent transformation experiments were performed with 3–5 composite plants analyzed per replica. (**B–C**) The eGFP and mCherry channels are shown in Green Fire Blue when isolated, with blue or yellow indicating low or high fluorescence levels, respectively. CD: cell division. En: endoreduplication. TD: terminal differentiation.

The online version of this article includes the following source data and figure supplement(s) for figure 1:

**Figure supplement 1.** Labeling patterns of selected Medicago histone H3 variants in transgenic roots.

**Figure supplement 1—source data 1.** Transcript levels of *H3.1* and *H3.3* genes in spot-inoculated roots (*Schiessl et al., 2019*) and statistical analyses.

**Figure supplement 1—source data 2.** Transcript levels of *H3.1* and *H3.3* genes in mature nodule regions (*Roux et al., 2014*) and statistical analyses.

in which nodule organogenesis is either abolished (*daphne-like*; *Liu et al., 2022*) or where nodules form but cortical infection threads are impaired in their progression (*nf-ya1-1*; *Laporte et al., 2014*). In the WT situation, the majority of cortical cells (90%) enabling transcellular passage until reaching the nodule primordium showed a reduced proliferation potential (low H3.1 content, n=41; *Figure 3A*, *Figure 3—figure supplement 1A–B*). A higher H3.1 content was occasionally observed in the last infected cell in the absence of nodule primordium organogenesis (NOD [(-)]; *Figure 3—figure supplement 1C–C″*). Absence of inner cortical cell divisions is also a characteristic feature of the *daphne-like* Medicago mutant (*Figure 3—figure supplement 2A–A″*), which lacks a remote cis-regulatory region in the *NIN* promoter perceiving cytokinin signals (*Liu et al., 2019b*). This non-nodulating mutant still produces WT-like root hair infections occasionally reaching outer cortical layers. In this context, the percentage of infected cortical cells visibly accumulating H3.1 in the euchromatin increased from 10% in the WT to almost 41% in the *daphne-like* mutant (n=44; *Figure 3A*, *Figure 3—figure supplement 2B–C*). This indicates that outer cortical cells hosting an infection thread lack extensive chromatin reorganization in the absence of sustained nodule organogenesis.

As a second mutant, we chose an *nf-ya1* null allele, producing smaller nodules that are additionally delayed and impaired in their development (*Laporte et al., 2014*; *Xiao et al., 2014*). Nodule primordia of the *nf-ya1-1* mutant showed reduced cell layers, with a lower frequency of cell divisions in the inner cortex (C4/5) and only a few anticlinal divisions in the middle layer (C3; *Figure 3—figure supplement 3A–B*; *Xiao et al., 2014*; *Lee et al., 2024*). Cortical infection threads appeared abnormally bulbous (*Figure 3A*, *3—figure supplement 3A*) and showed signs of early abortion (*Figure 3—figure supplement 3B*; *Laporte et al., 2014*; *Lee et al., 2024*). We observed that, even though the majority (76%) of crossed cortical cells were presumably arrested in a GAP phase (low H3.1 content, n = 54; *Figure 3—figure supplement 3A*), such pausing was not observed in about 24% of the inspected events. Here, infection threads were either blocked or delayed in the outer cortex, and the replicative H3.1 variant was retained (*Figure 3A*, *Figure 3—figure supplement 3B*). Although the last infected cortical cells showed significant nuclear enlargement in all three genetic backgrounds investigated (*Figure 3B*), failure to coordinate organogenesis and infection programs in the mutants resulted in an increased occurrence of infected cells accumulating more H3.1 than their direct neighbors (*Figure 3C*). Together, these data reveal that an appropriate attenuation in cell-cycle activity and associated chromatin changes in infected cells preferentially occur when progressing cortical infections and the formation of a nodule primordium are temporarily fully aligned.

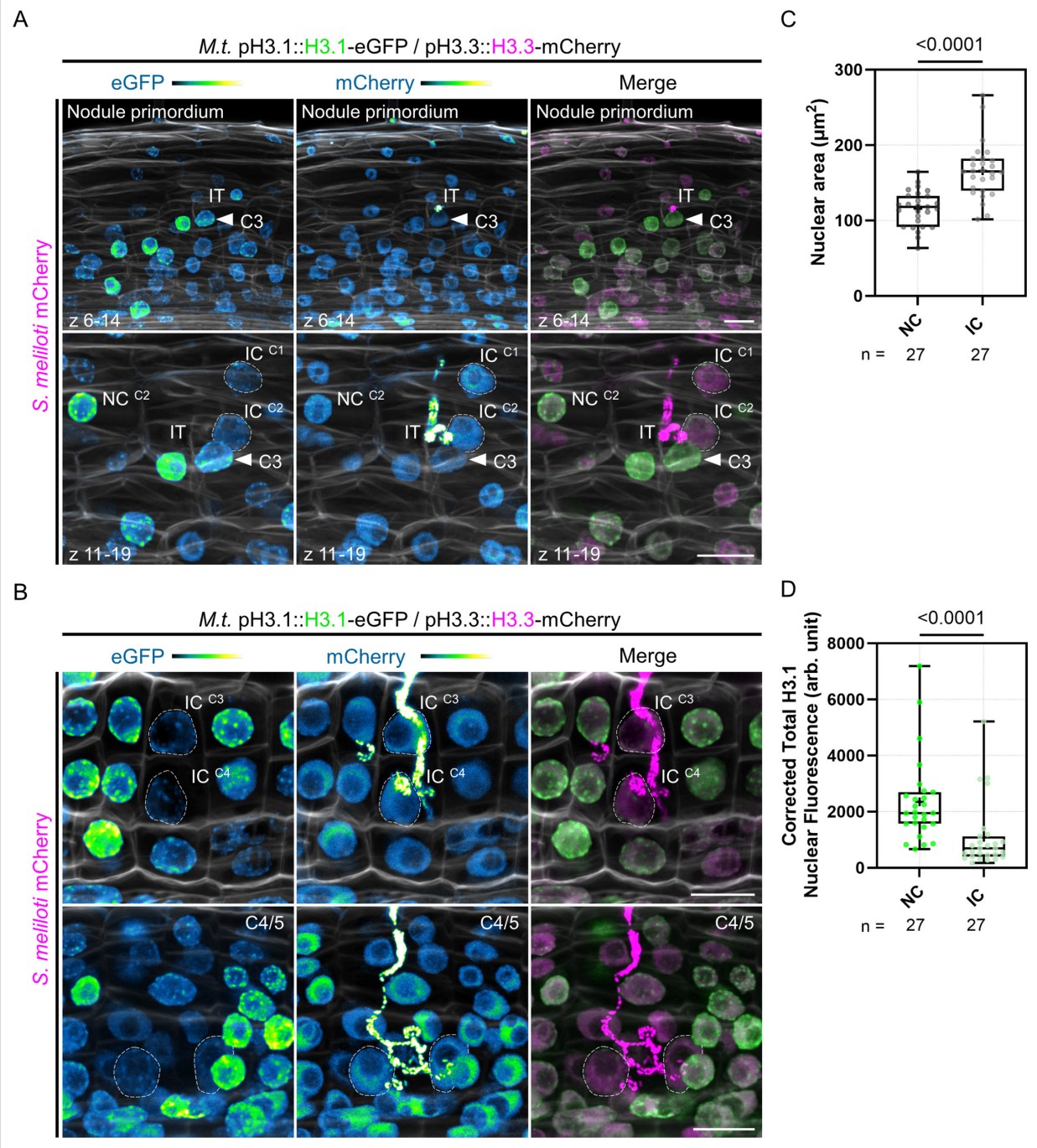

**Figure 2.** Individual reprogramming for infection includes large-scale chromatin rearrangements. (**A–B**) Confocal images of whole-mount WT roots expressing the pH3.1::H3.1-eGFP / pH3.3::H3.3-mCherry reporter and inoculated with mCherry-producing *S. meliloti* (8 dpi). Images are maximum intensity projections (eGFP: green; mCherry: magenta; Calcofluor white cell-wall staining: grayscale). The eGFP and mCherry channels are shown in Green Fire Blue when isolated. (**A**) Upper panels: view of inner and middle cortical regions of the early nodule primordium shown in lower panels. Lower panels: view of the outer cortical layers crossed by an infection thread (IT). Dashed lines demarcate nuclei of infected cells (IC) fully passed by a cortical infection thread in the first (C1) and second (C2) cortical layers. NC: neighboring cell. The filled arrowhead points to the nucleus of a host cell from the middle cortical layer (C3) which is penetrated by the IT. The upper and lower panels are from the same image. Scale bars: 20 μm. (**B**) Upper panels: dashed lines indicate nuclei of infected cells (IC) being passed (C4) or recently passed (C3) by a cortical infection thread. Lower panels: dashed lines demarcate nuclei of infected cells in the inner cortex (C4/5) of a nodule primordium with several cell layers. Scale bars: 20 μm. (**C–D**) Quantification of the nuclear area (**C**) and corrected total H3.1-eGFP nuclear fluorescence (**D**) at the equatorial plane in couples of neighboring (NC) and infected cells (IC) from the same cortical layer (C2 to C4, n=27; see Materials and methods for more details). Roots from two to seven composite plants with visible signs of inner cortical cell division (i.e. showing high H3.1-eGFP signal) were analyzed from three independent transformation experiments. All data points are

*Figure 2 continued on next page*

Figure 2 continued

shown and crosses indicate sample means. Differences were statistically significant (p-values <0.0001) using an unpaired t-test with Welch's correction (C) or a Mann-Whitney test (D).

The online version of this article includes the following source data and figure supplement(s) for figure 2:

**Source data 1.** Nuclear area and statistical analyses.

**Source data 2.** Corrected total H3.1 nuclear fluorescence and statistical analyses.

**Figure supplement 1.** A similar localization pattern and reduced H3.1 levels in infected cells are observed using a different *H3* gene.

## Cortical cells supporting transcellular infection control their commitment to cell division

In Medicago and other plants producing indeterminate nodules, a local control of mitotic activity could be especially important for rhizobia to timely cross the middle cortex, where sustained cellular proliferation gives rise to the nodule meristem (*Xiao et al., 2014*). To investigate in greater detail whether cells competent for transcellular infection are selectively kept away from transiting to mitosis, we first used a triple fluorescent sensor enabling us to track all cell-cycle phases in planta. Originally developed for *Arabidopsis*, the plant cell-cycle indicator (PlaCCI) combines a component of the pre-replication complex (CDT1a), the replicative histone variant (H3.1) and the N-terminal domain of a B-type cyclin (N-CYCB1;1) as G1-, S- and late G2-phase reporters, respectively (*Figure 4A*; *Desvoyes et al., 2020*; *Echevarría et al., 2021*). The N-terminal domain of CYCB1;1 contains a destruction box (D-box) ensuring a rapid turnover of the protein after one mitotic cell cycle. These cell-cycle markers are expressed at early stages of organ development but disappear in terminally differentiated cells (*Desvoyes et al., 2020*). When expressing the PlaCCI construct in transgenic Medicago roots, all three markers were successfully visualized in reactivated cortical cells upon nodule primordium formation (*Figure 4B–C*). Cortical cells passed by an infection thread kept low but detectable amounts of the *Arabidopsis* H3.1 variant, indicating that they transited through S-phase (*Figure 4C*). However, these cells stopped before accumulating substantial levels of N-CYCB1;1, in contrast to their direct neighbors. We also confirmed latter observations by using an *Arabidopsis* CYCB1;2-based fluorescent reporter (*Figure 5A*), labeling only cells that transit through G2/M (*Figure 5—figure supplement 1A*; *Weimer et al., 2016*). Based on these results and to adapt the PlaCCI reporter system to the study of legume-rhizobia interactions (*Nadzieja et al., 2019*; *Echevarría et al., 2021*), we designed an alternative version of the PlaCCI sensor (named PlaCCI v2) that includes the histone H3.1 variant from Medicago, functioning as a robust reporter of G2-arrested cells in composite plants. Observed in early nodule primordia, the PlaCCI v2 sensor enabled us to verify that the cells involved in trans-cellular infection presumably undergo a prolonged GAP phase without accumulating the B1;2 mitotic cyclin (*Figure 5—figure supplement 2A–C*). Together, these results confirmed a selective decrease in mitotic competence occurring on the cortical infection thread trajectory (*Yang et al., 1994*) as rhizobia pass the middle cortical layer (C3) to access colonizable cells in Medicago roots. Importantly, we did not observe the G1-phase marker accumulating in cells undergoing infection (*Figure 4B–D*, *Figure 5—figure supplement 2A–C*), favoring the hypothesis of a cell-cycle arrest and exit decision (*Figures 1A and 4A*) rather than a rapid switch to another endoreduplication round (*Desvoyes et al., 2019* – pre-print).

We next tested whether transcriptional reporters of a target gene necessary to execute cell division get selectively repressed on the path of a cortical infection. The late G2/M gene *KNOLLE* codes for a syntaxin protein specialized in angiosperm cytokinesis and is oppositely controlled when cells do (*Haga et al., 2007*) or shall not (*Takahashi et al., 2019*) commit to mitosis. The transcription of *KNOLLE* and other late cell-cycle genes (e.g. *CYCB1;2*) in *Arabidopsis* is primarily regulated by mitosis-specific activator (MSA) cis-elements present in their promoters that are targeted by activator- and repressor-type three Myb repeats (MYB3R) transcription factors (*Haga et al., 2007*; *Kobayashi et al., 2015*). Mutations in all three MSA core sequences present in the Medicago *KNOLLE* promoter significantly reduced the expression of a β-glucuronidase (*GUS*) reporter gene in *Nicotiana benthamiana* leaf cells in the presence of a hyperactive form of a MYB3R transcriptional activator (NtmybA2Δ630; *Figure 5—figure supplement 1B*; *Araki et al., 2004*), suggesting that the Medicago *KNOLLE* gene can be used as a *bona fide* late G2/M readout. Using the pKNOLLE::NLS-3xVenus construct, we found that the activation of the *KNOLLE* promoter was lower on the cortical infection thread trajectory in comparison

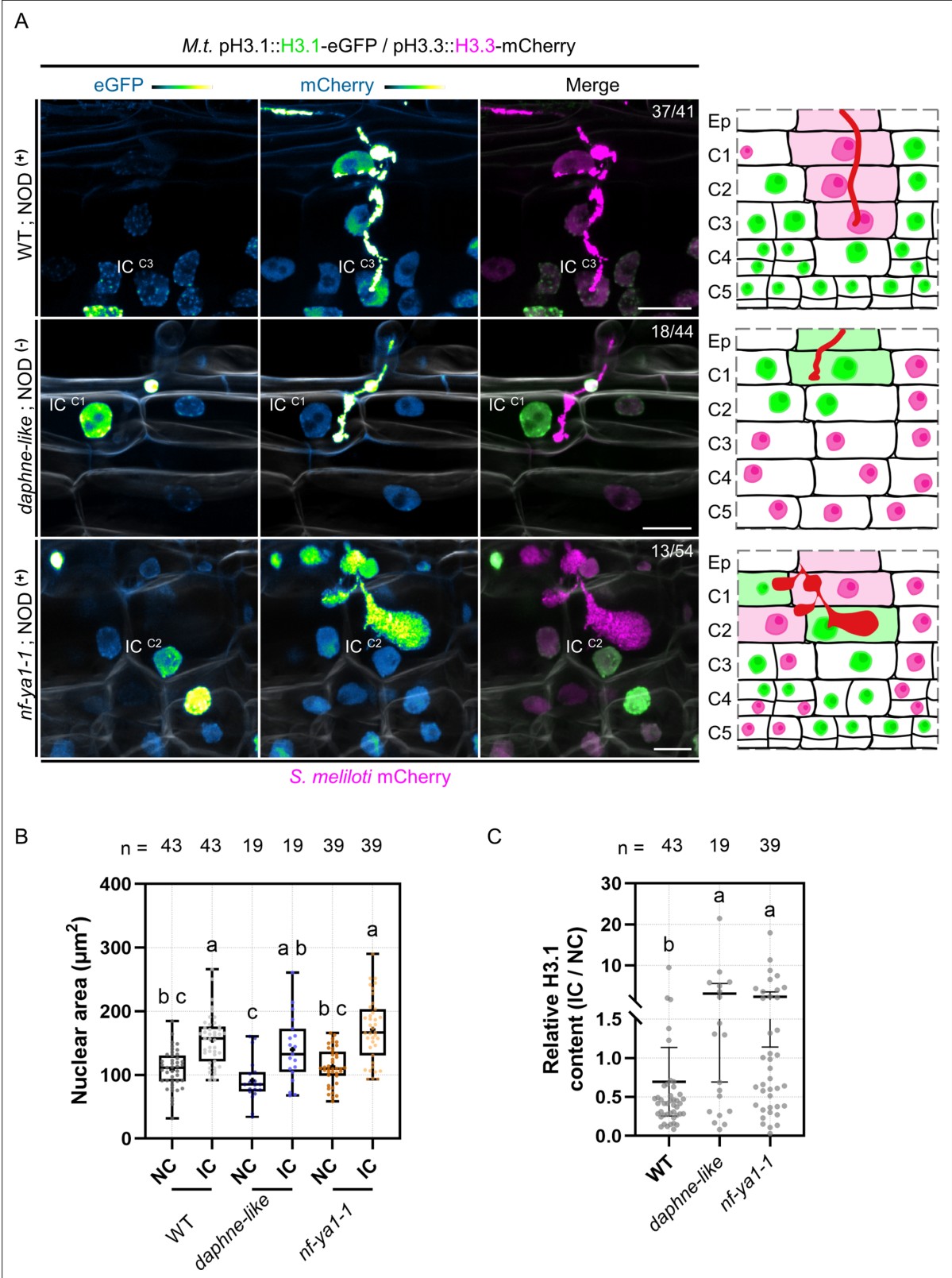

**Figure 3.** H3.1 eviction coincides with cortical infection thread progression. (**A**) Confocal images of whole-mount transgenic roots co-expressing H3.1-eGFP and H3.3-mCherry in three different genotypes: WT (11 dpi; upper panels), *daphne-like* (14 dpi; middle panels) and *nf-ya1-1* (12 dpi; lower panels), inoculated with mCherry-producing *S. meliloti*. WT and *nf-ya1-1* plants initiate nodule formation (NOD[+]), whereas the *daphne-like* mutant is non-nodulating (NOD[-]). Numbers indicate the frequencies of observation of a low (WT) or high (*daphne-like*, *nf-ya1-1*) H3.1-eGFP content in the last

*Figure 3 continued on next page*

*Figure 3 continued*

infected cortical cell. Corresponding schematic representations depict cellular proliferation or endocycling activities indicated by nuclear H3.1 levels (low: magenta; high: green) on the cortical infection thread trajectory (C1 to C3; diffuse magenta or green coloring) and in inner cortical layers (C4 to C5; see also *Figure 3—figure supplements 1–3*). Rhizobia inside infection threads are depicted in red. Confocal images show merged fluorescent channels (eGFP: green; mCherry: magenta; Calcofluor white cell-wall staining: grayscale in the middle and lower panels). The eGFP and mCherry channels are shown in Green Fire Blue when isolated. Scale bars: 20 μm. IC: infected cell. Ep: epidermis. (**B–C**) Quantification of the nuclear area (**B**) and the relative H3.1-eGFP content (IC/NC; corrected total nuclear fluorescence in arbitrary units) (**C**) at the equatorial plane in couples of neighboring (NC) and infected cells (IC) from the same cortical layer in WT (C1 to C4), *daphne-like* (C1 to C2) and *nf-ya1-1* (C1 to C4) genetic backgrounds (n=43, 19, and 39 nuclei, respectively). Different letters indicate statistically significant differences according to a Kruskal-Wallis test followed by Dunn's multiple comparisons test. (**B**) All data points are shown and crosses indicate sample means. (**C**) All data points are shown and horizontal bars indicate sample means with 95% confidence interval. Ten to 21 composite plants (8–14 dpi) from two (*daphne-like*, *nf-ya1-1*) to 7 (WT) independent transformation experiments were analyzed.

The online version of this article includes the following source data and figure supplement(s) for figure 3:

**Source data 1.** Nuclear area and statistical analyses.

**Source data 2.** Relative H3.1 corrected total nuclear fluorescence and statistical analyses.

**Figure supplement 1.** Infected outer cortical cells do not evict H3.1 when infection and organogenesis processes are uncoupled.

**Figure supplement 2.** Sustained cell-cycle activity in the outer cortical layers is a characteristic of the non-nodulating *daphne-like* mutant.

**Figure supplement 3.** Maintenance of H3.1 in infected cells in the *nf-ya1-1* mutant coincides with a suboptimal state of cell division and progression of the cortical infection thread.

to neighboring, proliferating cells from nodule primordia (*Figure 5B*). An equivalent construct, ensuring the timely elimination of fluorescent reporters after one canonical cell cycle (pKNOLLE::D-box-3xVenus-NLS; *Figure 5—figure supplement 1C*), corroborated a selective control of the G2/M transition in cells undergoing transcellular infection. Combining our observations of enlarged nuclei (*Figure 2C*), the absence of a G1 marker, and the reduced H3.1 levels and expression of mitotic genes (*Figures 2D and 4B–D* and *Figure 5A–B*), we conclude that the initial cortical cells along the infection thread trajectory conduct a single cell-cycle round and stay in a post-replicative (i.e. G2) phase as long as they are crossed.

We further took advantage of the fact that the PlaCCI sensor provides a full view on cell-cycle phase progression in root tissues, in *Arabidopsis* (*Desvoyes et al., 2020*) and Medicago (this study). CDT1a- and CYCB1;1-associated signals in the root tip enabled us to identify 2C (i.e. G1) and 4C (i.e. G2) nuclei, respectively (*Figure 5—figure supplement 3A*) – where 2C and 4C chromatin values correspond to the DNA content of a basic diploid genome. Measuring the area of nuclei exhibiting the CYCB1;1-YFP signal in cortical layers of the root apical meristem (root tip) or the nodule primordium revealed that they were on average 1.57 or 1.63 times larger than the ones producing CDT1a-eCFP, respectively (*Figure 5—figure supplement 3B*). This is in agreement with an increased DNA content as cells transit from the G1- to the late G2-phase. Surprisingly, proliferating nodule cells contained significantly wider nuclei than their diploid counterparts in the proliferation zone of the root apical meristem (*Figure 5—figure supplement 3B*), suggesting a higher ploidy level early on. These results lead to the intriguing possibility that reactivated inner cortical cells in Medicago enter the mitotic cell cycle with an increased amount of genetic material.

## Proliferation of endopolyploid cortical cells is a distinctive feature of Medicago root nodules

Our data suggest that the dividing cortical cells contributing to nodule primordium formation are polyploid – that is they contain more than the two genome copies of a diploid (2n) cell, where 'n' refers to the number of separate chromosomes. To assess this further, we identified and used the centromere-specific histone H3 variant (CENH3) from Medicago as a marker of individual chromosomes. Such fluorescently tagged CENH3 proteins have already been successfully applied for chromosome quantification in planta (*Lermontova et al., 2006*; *De Storme et al., 2016*). Aligning the Medicago CENH3 amino acid sequence (G8A083) with characterized CENH3 sequences from *L. japonicus*, *Arabidopsis*, *H. vulgare*, and *Oryza sativa* revealed the presence of canonical CENH3 features (*Figure 6—figure supplement 1*), including a variable N-terminal tail and the centromere targeting domain (CATD). When expressing a mCitrine-CENH3 fusion protein under the control of

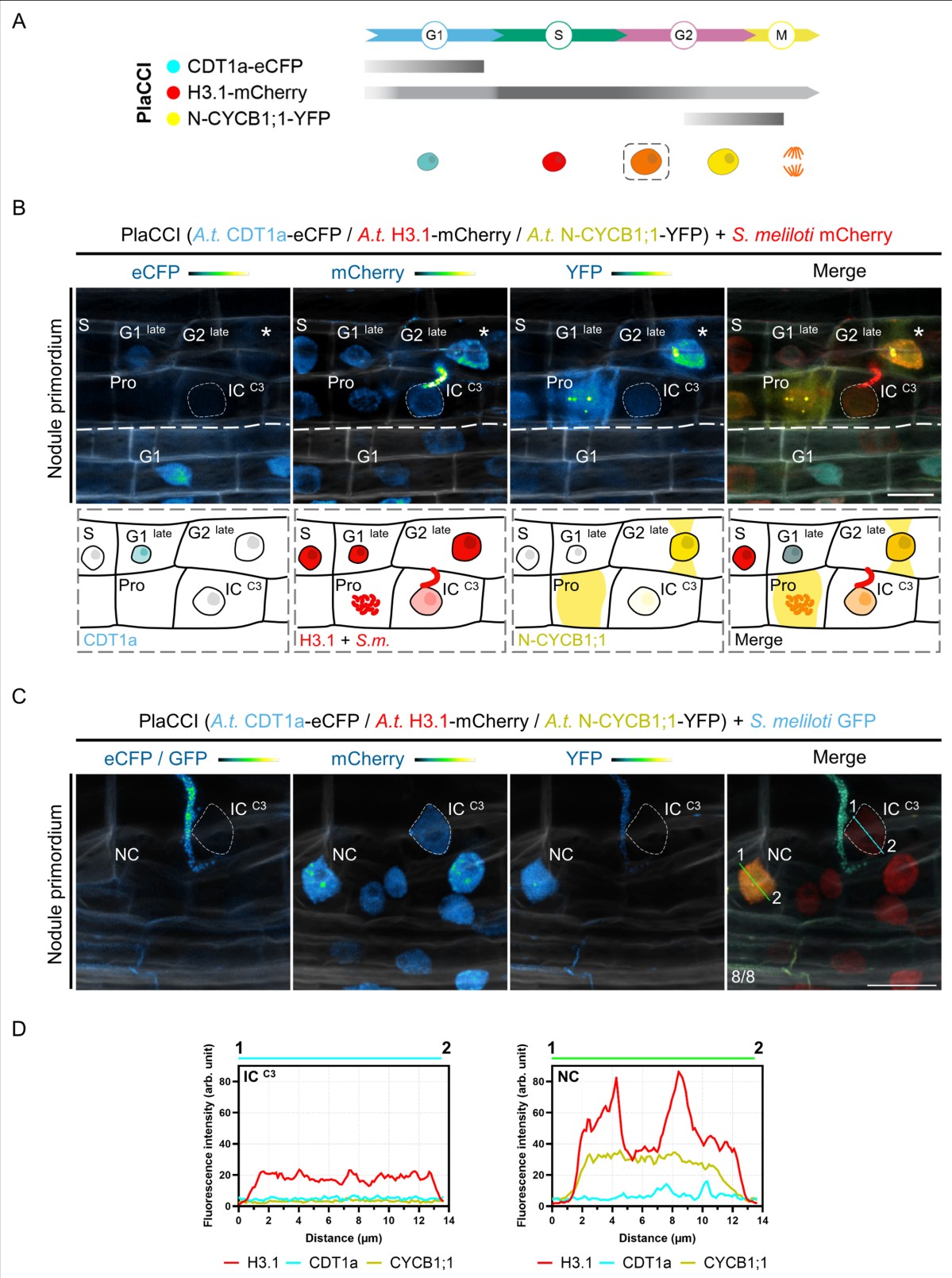

**Figure 4.** The reduction in the proliferative potential of infected cortical cells is supported by the *Arabidopsis* PlaCCI reporter. (**A**) Schematic representation of *Arabidopsis* CDT1a-eCFP, H3.1-mCherry and N-CYCB1;1-YFP distribution (horizontal bars) and fluorescence intensity (gray saturation) throughout the different cell-cycle phases (adapted from ***Echevarría et al., 2021***; see also ***Desvoyes et al., 2020***). CDT1a-CFP accumulates in G1 and is rapidly degraded during the G1/S transition (blunt end bar). H3.1 is predominantly expressed during S-phase and incorporated during DNA

*Figure 4 continued*

replication. N-CYCB1;1-YFP is present in late G2 and mitotic cells and is completely degraded in anaphase (blunt end bar). Nuclei of increasing size and DNA content are colored according to their CDT1a (cyan), H3.1 (red), and N-CYCB1;1 (yellow) content in/around decondensed chromatin (G1, S, G2) or condensed chromosomes (M). H3.1 levels decrease in differentiating cells (dashed line box). (**B–C**) Confocal images of whole-mount Medicago WT roots expressing the *Arabidopsis* PlaCCI reporter in nodule primordia at 7 dpi with mCherry- (**B**) or GFP-producing *S. meliloti* (**C**). Images show merged fluorescent channels (eCFP / GFP: cyan; mCherry: red; YFP: yellow; Calcofluor white cell-wall staining: grayscale). Individual channels (eCFP / GFP, mCherry, YFP) are shown in Green Fire Blue when isolated. (**B**) Upper panels: the thin dashed line demarcates the nucleus of an infected cell (IC) from the C3 layer penetrated by an infection thread. Cell-cycle phases of non-infected neighboring cells are indicated. The diffusion of the N-CYCB1;1-YFP marker outside the nucleus allows visualization of the phragmosome transvacuolar bridge formed in preparation for mitosis (stars). A cytoplasmic accumulation of the same marker (YFP channel) is visible after nuclear envelope breakdown in a prophase (Pro) cell showing condensing chromosomes (mCherry channel). Scale bar: 20 µm. Lower panels: schematic representations illustrating the different phases of the cell cycle visible above the thick dashed line, revealed by the PlaCCI reporter in the upper panels. Cells engaged in recurrent cell division cycles (G1 $^{late}$, S, G2 $^{late}$, prophase) maintain a high level of H3.1 (red). The infected cell (IC) is distinguished by the absence of visible CDT1a-CFP (cyan), a reduced level of H3.1, and a very low level of N-CYCB1;1-YFP (yellow) compared to neighboring cells. (**C**) The dashed line demarcates the nucleus of an infected cell (IC) from the C3 layer being passed by a cortical infection thread. Numbers indicate the frequencies of observation of the absence of CDT1a- or N-CYCB1;1-associated signals in the nucleus of the last infected cortical cell. Scale bar: 20 µm. (**D**) Fluorescence intensity profiles of CDT1a-, H3.1-, and N-CYCB1.1-associated signals along the cyan and green transects shown in (**C**).

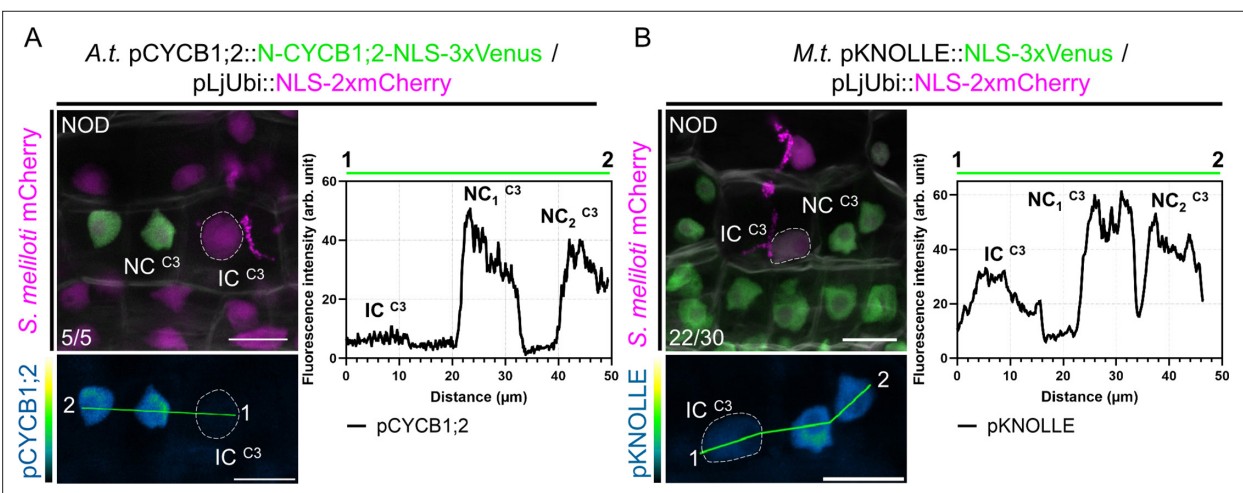

**Figure 5.** A tight control over host cells' mitotic commitment enables passage of the future nodule meristem. (**A**) Left panels: confocal images of a whole-mount WT root co-expressing a destabilized triple-Venus nuclear reporter driven by the *Arabidopsis CYCB1;2* promoter (pCYCB1;2::N-CYCB1;2-NLS-3xVenus) and a nuclear-localized tandem-mCherry as a transformation marker, 12 dpi with mCherry-producing *S. meliloti*. Numbers indicate the frequencies of observation of the absence of the triple-Venus reporter signal in the nucleus of the last infected cortical cell. Two independent transformation experiments were performed with three to four composite plants analyzed per replica. (**B**) Left panels: confocal images of a whole-mount WT root co-expressing a transcriptional reporter of Medicago *KNOLLE* driving a nuclear-localized triple-Venus (pKNOLLE::NLS-3xVenus) together with a nuclear-localized tandem-mCherry as a transformation marker, 12 dpi with mCherry-producing *S. meliloti*. Numbers indicate the frequencies of observation of nodule primordia where the triple-Venus reporter signal is kept comparably low on the cortical infection thread trajectory. A total of 12 composite plants from two independent transformation experiments were analyzed. (**A–B**) Images show merged fluorescent channels (Venus: green; mCherry: magenta; Calcofluor white cell-wall staining: grayscale in the upper panels). The Venus channel is shown in Green Fire Blue when isolated in lower panels. The dashed lines indicate nuclei of infected cells (IC) from the C3 layer being passed by a cortical infection thread. The fluorescence intensity profiles of pCYCB1;2 and pKNOLLE reporter-associated signals along the green transects are shown in the corresponding right panels. NC: neighboring cell. NOD: nodule primordium. Scale bars: 20 µm.

The online version of this article includes the following source data and figure supplement(s) for figure 5:

**Figure supplement 1.** *KNOLLE* and *CYCB1;2* transcriptional reporters highlight individual cells preparing for mitosis in Medicago.

**Figure supplement 1—source data 1.** GUS activity and statistical analysis.

**Figure supplement 2.** A PlaCCI sensor optimized for identifying cells competent for infection highlights their pre-mitotic arrest during transcellular passage of the middle cortex.

**Figure supplement 3.** Proliferating cells in the nodule primordium exhibit significantly larger nuclei compared to meristematic root tip cells.

**Figure supplement 3—source data 1.** Nuclear area and statistical analysis.

its own promoter (pCENH3::mCitrine-CENH3), CENH3 labeled up to 16 nuclear foci and localized exclusively at the presumed centromeres during mitosis (*Figure 6A and B*) in diploid root tip cells. As the signal was rather weak, we scored centromeric foci in roots constitutively expressing the protein (pLjUbi::mCitrine-CENH3; *Figure 6A*). Here, a reliable signal was observed in an increased proportion of nuclei (82%) displaying not more than 14–16 mCitrine-CENH3 foci (versus only 55% when *CENH3* was expressed under its endogenous promoter; n=55; *Figure 6B*) – reflecting the base ploidy in *Medicago truncatula* with 2n = 16. We therefore used the pLjUbi::mCitrine-CENH3 reporter construct for our further analyses.

In a first set of experiments, we used CENH3 to discriminate between polyploid cells undergoing endomitosis or endocycling. In endomitosis, entire chromosome replication and partial mitotic progression result in a doubling of the chromosome / centromeric foci number (*Iwata et al., 2011*; *De Storme et al., 2016*). By contrast, endocycling cells undergo repeated rounds of DNA synthesis without segregating the newly replicated parts (*Edgar et al., 2014*). Here, copied chromosome arms do not condense and spatial separation of sister kinetochores does not occur, so that the number of compact CENH3 signals corresponds to the initial number of chromosomes (up to 8C; *Figure 6—figure supplement 2A*; *Lermontova et al., 2006*). As the occurrence of endocycling cells has been previously reported in tissues infected by AM fungi, we first tested the CENH3-reporter in roots inoculated with *Rhizophagus irregularis*. Indeed, cortical cells entered by the fungus displayed large nuclei with no more than 16 labeled foci (*Figure 6—figure supplement 2B*), supporting their endocycling nature (*Carotenuto et al., 2019a*; *Lermontova et al., 2006*). Up to 16 CENH3 foci were also observed in diploid cortical cells from the root tip and pericycle cells from lateral root primordia (*Figure 6—figure supplement 2C*). However, the majority of nodule cells exhibited between 26 and 32 fluorescent centromeric foci (75.5% in nodule primordia from stage IV to VI, n = 106; *Figure 6—figure supplement 2B–C*) and a visibly increased number of mitotic chromosomes (*Figure 6—figure supplement 2B*). These observations strongly support previously published data proposing that proliferating cortical cells giving rise to indeterminate root nodules are tetraploid (4n = 32 in *Medicago truncatula*; *Torrey and Barrios, 1969*).

We next tested whether endopolyploid cells are already present at an earlier stage of nodule development, where cell division events are confined to the pericycle, the inner cortical cells (C4, C5) and occasionally occur in the third (C3) and second (C2) cortical layers (*Figure 6C*; *Xiao et al., 2014*). Indeed, cells originating only from cortical cell divisions (C2 to C5) showed an increased number of chromosomes (*Figure 6D–E*) compared to recently divided cells originating from the pericycle (*Figure 6E*). Notably, cells performing anticlinal divisions in the middle cortex (*Figure 6D*, left panel) that will later be crossed by the infection thread (*Figure 6E*, violet triangle) were also polyploid. From these observations, we conclude that a key feature of cortical cells in Medicago nodules is their ability to segregate a higher number of chromosome copies – corresponding to 8C in a tetraploid cell entering mitosis.

## Infected cells within the nodule primordium selectively halt their preparations for mitosis

In the next set of experiments, we studied to which extent the cells positioned on the infection thread trajectory (C3 to C4/5) prepare for mitosis. For this, we made use of the pre-mitotic loading characteristics of CENH3 to the centromeres, which mainly occurs during the G2 phase and prior to the spatial separation of sister kinetochores (*Figure 7A*; *Lermontova et al., 2007*; *Lermontova et al., 2011*; *Lermontova et al., 2006*). In proliferating populations of the root tip (*Figure 7—figure supplement 1A–B*) and nodule primordia (*Figure 7A–B*), late G2 cells contained large nuclei where the majority of sister kinetochores was split and appeared as twin foci. We further noticed that centromere separation in Medicago is a gradual rather than a simultaneous process, as root tip cells with comparable nuclear areas often exhibited an intermediate number of fluorescent doublets (*Figure 7—figure supplement 1A*) of similar appearance and length on a single focal plane (*Figure 7—figure supplement 1B–D*). To address the question to which extent cortical cells that are competent for infection can commit to nuclear division, we assessed the gradual formation of sister centromeres. Although the nuclear size of infected C3 cells suggested an 8C DNA content, as found in late G2-phase cells of the nodule tissue (*Figure 7B and C*), none of these infected cells showed the typical CENH3 twin foci formed when sister kinetochores split (n = 13; *Figure 7B and D*). Such control over mitotic commitment was tightly

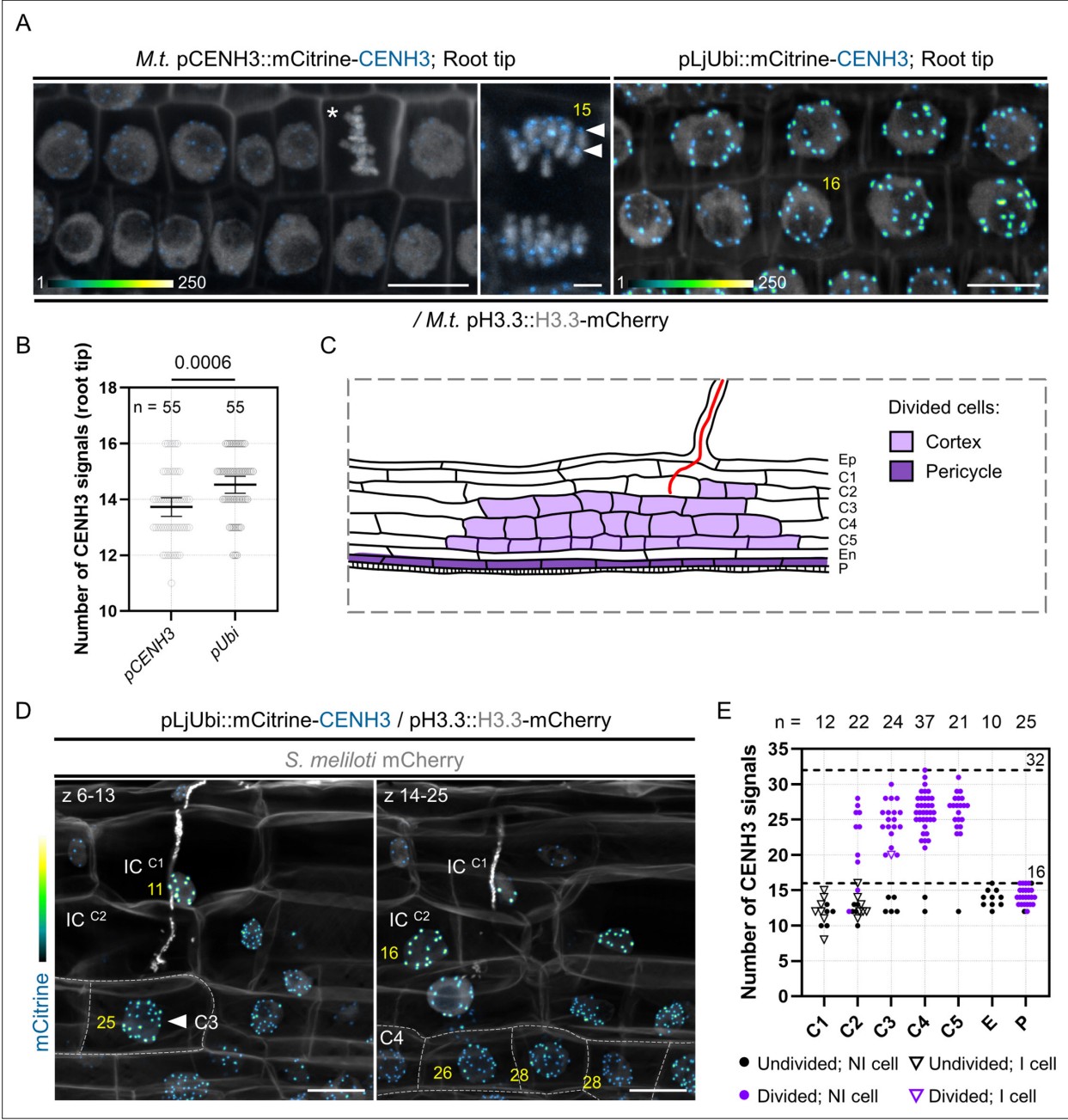

**Figure 6.** Dividing cortical cells initiating Medicago nodule primordium formation are tetraploid. (**A**) Confocal images of whole-mount WT transgenic roots expressing mCitrine-CENH3 under the control of native (*M.t.* pCENH3; left and middle panels) or constitutive promoters (pLjUbi; right panel). Simultaneous expression of H3.3-mCherry enables the recognition of condensed (star) and segregating chromosomes (middle panel). mCitrine-CENH3 labels the centromeric region of individual chromosomes (filled arrowheads). The number of CENH3-labeled foci determined across image stacks is indicated in yellow. Images show merged fluorescent channels (mCitrine: Green Fire Blue; mCherry and Calcofluor white cell-wall staining: grayscale). In the Green Fire Blue color scheme, blue or yellow indicate low or high fluorescence levels, respectively (min. to max.=1–140 in the middle panel). Scale bars: left and right panels = 10 μm; middle panel = 2 μm. (**B**) Quantification of the number of centromeric signals in transgenic root tips expressing mCitrine-CENH3 under native (pCENH3) or constitutive (pLjUbi) promoters (n=55 nuclei). All data points are shown and are from two (pLjUbi) to three (pCENH3) independent experiments with seven composite plants analyzed per construct. Horizontal bars indicate sample means with 95% confidence interval. Differences were statistically significant (p-value = 0.0006) using a Mann-Whitney test. (**C**) Schematic representation of an early nodule primordium where anticlinal cell divisions occurred in the pericycle (P; dark violet) and cortical cells (C2 to C5; light violet). Rhizobia inside the infection thread are depicted in red. Ep: epidermis. En: endodermis. (**D**) Maximum intensity projections of an early nodule primordium at the developmental stage represented in (**C**), formed in a WT root expressing the pLjUbi::mCitrine-CENH3 /pH3.3::H3.3-mCherry construct at 7 dpi with mCherry-producing *S. meliloti*. Dashed lines indicate the contours of divided cells in the inner (C4) and middle (C3) cortical layers. The number of CENH3-labeled foci,

*Figure 6 continued on next page*

Figure 6 continued

determined across image stacks and indicated in yellow, is: 11 and 16 in undivided, infected cells (IC) from the outer cortical layers (C1; left panel and C2, right panel) and 25–28 in divided, uninfected cells from the C3 (filled arrowhead; left panel) and C4 layers (right panel). Images show merged fluorescent channels (mCitrine: Green Fire Blue; mCherry and Calcofluor white cell-wall staining: grayscale). Scale bars: 20 μm. (**E**) Quantification of the number of centromeric signals in early nodule primordia at the developmental stage depicted in (**C**), formed in inoculated WT roots (7–12 dpi) constitutively expressing mCitrine-CENH3. The number (n) of analyzed nuclei in each cell layer is indicated on the top. E: endodermis. P: pericycle. All data points are shown with black or violet symbols indicating undivided or divided cells, respectively. NI: non-infected cell (discs). I: infected cell (triangles). Horizontal dotted lines are positioned at y=16 and y=32, corresponding to diploid (2n=16) or tetraploid (4n=32) cellular states in *Medicago truncatula*. Data are from two independent experiments with seven nodule primordia from five composite plants analyzed.

The online version of this article includes the following source data and figure supplement(s) for figure 6:

**Source data 1.** Number of CENH3 signals (root tip) and statistical analysis.

**Source data 2.** Number of CENH3 signals (early nodule primordia).

**Figure supplement 1.** Multiple sequence alignment of plant CENH3 proteins.

**Figure supplement 2.** Somatic centromere labeling identifies a doubling of the chromosome number as a typical feature of nodule cortical cells.

**Figure supplement 2—source data 1.** Number of CENH3 signals (root tip, lateral root primordia, nodule infection zone).

maintained in the majority of infected, C4/5-derived cells (75.5%; n = 53; *Figure 7D*) which did not display any visible sign of centromere separation across image stacks (*Figure 7B*), despite a presumably duplicated genomic DNA content (*Figure 7C*). A smaller proportion of the recently infected cells (20.7%) partially prepared for chromosome segregation and showed up to three fluorescent doublets (*Figure 7D*) of comparable lengths to those measured in G2-phase progressing cells (*Figure 7E*), whereas only two cells (3.8%) with internalized bacteria got closer to nuclear and likely to cell division (*Figure 7B and D*). Together, by combining several marker strategies based on histone H3 dynamics and on bona fide mitotic reporters, we conclude that most middle (C3) and inner (C4/5-derived) cortical cells competent for intracellular infection enter a post-replicative, G2-type phase without proceeding toward M-phase. Endopolyploid cortical cells hosting infection threads have presumably gained the ability to execute a differentiation program rather than entering cell division at the end of G2.

## The symbiosis-specific NF-YA1 subunit holds potential to modulate the G2/M transition

Our data suggest that cell-cycle progression and bacterial infection are highly coordinated, although the precise molecular players able to redirect cell-cycle decisions while promoting cortical infections remain to be identified. Prominent candidates are NF-YA1 proteins that were evolutionary recruited to RNS, where they sustain both organogenesis and intracellular infection in a range of nodulating legume and non-legume species (*Combier et al., 2006*; *Laporte et al., 2014*; *Xiao et al., 2014*; *Hossain et al., 2016*; *Bu et al., 2020*). NF-YA1 plays an unambiguous role as a positive regulator of cell divisions (*Soyano et al., 2013*) but was also recently shown to function in the early differentiation and specification of nodule cells in Medicago (*Lee et al., 2024*). This prompted us to assess whether NF-YA1 could act as a switch at mitotic entry. To test this, we used WT NF-YA1 and a mutated variant lacking a functional DNA-binding domain (mutDBD) as eGFP fusion proteins. As expected, both variants localized to the nucleus when being expressed in *N. benthamiana* leaf epidermal cells (*Figure 8A*; *Laloum et al., 2014*). To foster cell cycle re-entry in these differentiated cells, we ectopically expressed the *Arabidopsis* D-type cyclin *CYCD3;1* in mature leaves as described in the recently developed 'cell division-enabled leaf system' (CDELS; *Xu et al., 2020*). As these re-activated cells commit to mitosis and perform cytokinesis (*Figure 8B*), they also enable the activation of promoters containing MSA cis-elements (*Xu et al., 2020*) as found in the Medicago *KNOLLE* promoter (*Figure 5—figure supplement 1B*). We confirmed this by expressing eGFP-KNOLLE from its endogenous promoter in the absence and presence of *Arabidopsis* CYCD3;1. The KNOLLE fusion protein was only detected when being co-expressed with CYCD3;1 using immunoblot analyses (*Figure 8—figure supplement 1A*) and labeled newly formed cell-division planes (*Figure 8—figure supplement 1B*). This is consistent with previous reports showing that the syntaxin KNOLLE accumulates only in dividing cells. We also confirmed the specific activation of the Medicago *KNOLLE* promoter (pKNOLLE::GUS) in CDEL samples ectopically expressing CYCD3;1 by using a fluorometric GUS assay (*Figure 8C*). Next, we

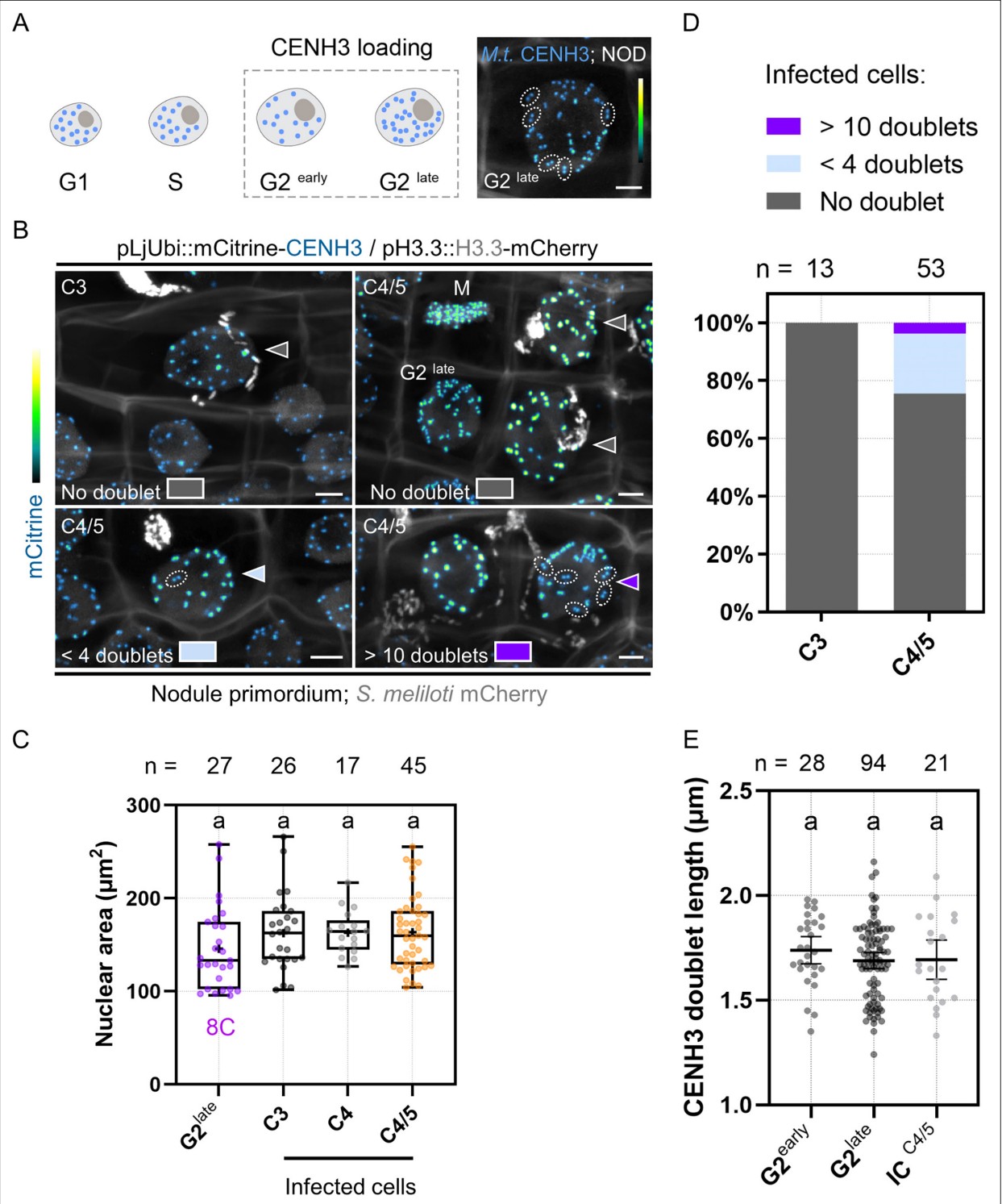

**Figure 7.** Primary infected nodule primordium cells do not reach full competence for chromosome segregation. (**A**) Schematic representation of CENH3 deposition at centromeres (blue dots) throughout the different pre-mitotic (G1, S, G2$^{early}$, and G2$^{late}$) cell-cycle phases in plants. Nuclei of increasing size and DNA content are depicted as ovals. The dashed line box indicates the timing of CENH3 loading at replicated centromeres, occurring during G2 before sister kinetochore (blue doublets) split in preparation for mitosis (**Lermontova et al., 2007**). Right panel: confocal image of a late G2-phase cell observed in a nodule (NOD) in an inoculated WT transgenic root (12 dpi) constitutively expressing mCitrine-CENH3. The majority of fluorescent signals appear as doublets (encircled by dotted lines) corresponding to sister kinetochores. Scale bar: 5 µm. (**B**) Confocal images of whole-mount WT roots expressing the pLjUbi::mCitrine-CENH3 /pH3.3::H3.3-mCherry construct and inoculated with mCherry-producing *S. meliloti* (7–12 dpi). Filled arrowheads point to nuclei of infected nodule primordium cells. Dotted lines encircle mCitrine-CENH3 doublets appearing as twin spots on the same

*Figure 7 continued on next page*

Figure 7 continued

focal plane. Neighboring cells in late G2 or undergoing mitosis (M) are indicated. The presence of CENH3-labeled twin spots was assessed across image stacks in the nuclei of infected cells from the middle (C3) and inner (C4/5) cortical layers and color-coded as follows: gray = no doublet (upper panels); light blue = 1–3 doublets (lower left panel); dark violet = 11 or more doublets (lower right panel). Scale bars: 5 μm. (**A–B**) Images show merged fluorescent channels (mCitrine: Green Fire Blue; mCherry and Calcofluor white cell-wall staining: grayscale). (**C**) Quantification of the nuclear area at the equatorial plane in cells being passed by a cortical infection thread (infected cells) in nodule primordia formed by WT transgenic roots inoculated with *S. meliloti* (7–12 dpi). The number (n) of analyzed nuclei in each cell layer (C3 to C4/5) is indicated on the top. The 8C chromatin value is given for uninfected nodule primordium cells in late G2 showing close to 32 mCitrine-CENH3 doublets as identified in (**A**) and (**B**). All data points are shown, and crosses indicate sample means. Differences were not statistically significant according to a Kruskal-Wallis test followed by Dunn's multiple comparisons test. At least nine composite plants from two independent experiments were analyzed. (**D**) Quantification of infected cells in C3 (n=13) and C4/5 (n=53) cortical layers in nodule primordia, showing no doublets (gray), 1–3 doublets (light blue) or more than 11 doublets (dark violet) labeled by mCitrine-CENH3. Nine composite plants from two independent experiments were analyzed. (**E**) Quantification of the length of mCitrine-CENH3 doublets appearing as twin spots on the same focal plane in early G2 (n=12), late G2-phase cells (n=16), and infected cells (IC) from the C4/5 cortical layer (n=13) in nodule primordia. n=28 (G2$^{early}$), 94 (G2$^{late}$), and 21 (IC$^{C4/5}$) doublets. Horizontal bars indicate sample means with 95% confidence interval. Differences were not statistically significant according to a Kruskal-Wallis test followed by Dunn's multiple comparisons test. All data points are shown and are from five to seven composite plants from two independent transformation experiments.

The online version of this article includes the following source data and figure supplement(s) for figure 7:

**Source data 1.** Nuclear area and statistical analysis.

**Source data 2.** CENH3 doublet length and statistical analysis.

**Figure supplement 1.** CENH3 distribution supports the identification of G2-phase cells.

**Figure supplement 1—source data 1.** Nuclear area and statistical analyses.

**Figure supplement 1—source data 2.** CENH3 doublet length and statistical analyses.

tested the impact of NF-YA1 on the activity of promoters driving cell-cycle-dependent gene transcription in the CDEL samples. While the presence of NF-YA1 did not alter *GUS* expression when driven from the histone *H4* promoter (pH4::GUS) that is active during DNA synthesis (*Figure 8C–D*; *Yang et al., 1994*), it significantly repressed pKNOLLE::GUS activation (*Figure 8E*). This effect was abolished when using the mutated NF-YA1$^{mutDBD}$ variant (*Figure 8E*). Interestingly, the repressive effect of NF-YA1 was stronger compared to the one observed when using a MYB3R transcriptional repressor from Medicago (*Figure 8—figure supplement 2A–B*) known to halt the expression of late G2/M genes (*Haga et al., 2007*; *Takahashi et al., 2019*). These results suggest that the NF-YA1 DNA-binding subunit prevents an unscheduled entry into mitosis in de-differentiating leaf cells, despite the action of ectopically expressed CYCD3;1.

We then tested whether such an effect could be observed at the cellular level in the CDEL system. For this, we ectopically co-expressed non-fluorescently tagged CYCD3;1 and NF-YA1 and additionally transformed these leaves with fluorescently tagged Medicago H3.3 and KNOLLE proteins, expressed from their native promoters (*Figure 8—figure supplement 1B*). While the constitutive presence of NF-YA1 did not prevent cell division events per se, a significantly lower proportion of transformed cells produced detectable levels of KNOLLE (*Figure 8—figure supplement 1C*), indicating that fewer pavement cells engaged in mitosis at the investigated experimental timepoint. Cells visibly accumulating the KNOLLE protein still progressed through M-phase in the presence of NF-YA1 (*Figure 8—figure supplement 1B*), although a smaller proportion of them fully completed cell division at 64 hours when compared to the control samples (*Figure 8—figure supplement 1D*). From these data, we conclude that NF-YA1 is able to control mitotic entry in de-differentiating, endoreduplicated cells, at least in the CDEL system.

## NF-YA1 acts at the interface between control of cell division and intracellular infection competence in the Medicago root cortex

According to our observations, access to colonizable cells *via* transcellular infection implies a local decision to exit the cell cycle. Given that NF-YA1 holds potential to control the G2/M transition when expressed in a heterologous system (*Figure 8E*, *Figure 8—figure supplement 1B–D*), we sought to test the consequences of its accumulation along the infection thread trajectory. Consequently, and to avoid potential developmental side effects by constitutively and ectopically overexpressing *NF-YA1*, we first checked the cellular activation profile of two symbiosis-specific genes recruited during infection thread formation and growth, *ENOD11* (*Journet et al., 2001*) and *NPL* (*Xie et al., 2012*). Their

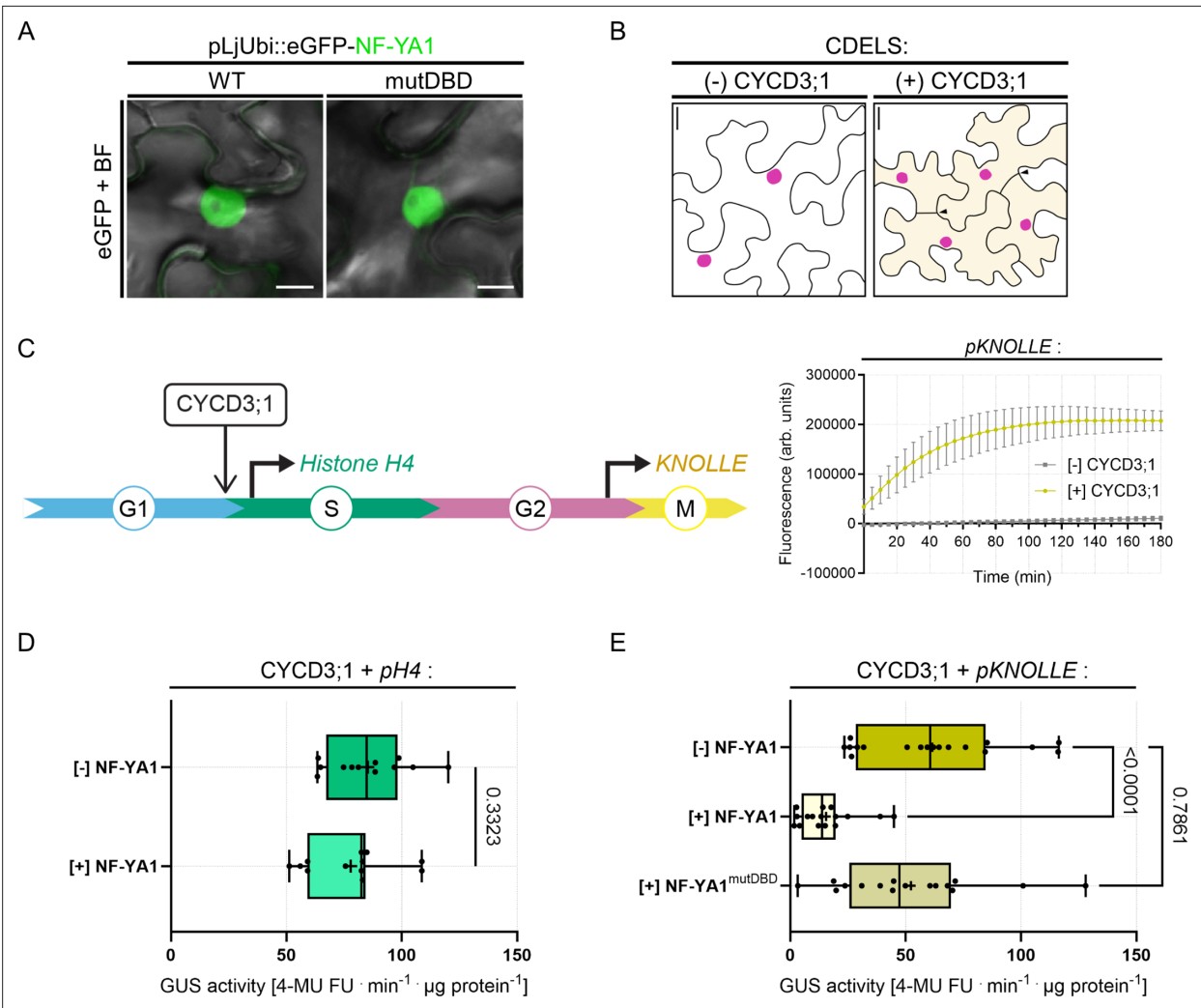

**Figure 8.** NF-YA1 specifically reduces the expression of a G2/M gene in a cell-division enabled system. (**A**) Live-cell confocal images of *N. benthamiana* epidermal cells ectopically expressing eGFP-NF-YA1 variants: WT (left panel) or impaired in DNA recognition (mutDBD; right panel). Both fusion proteins accumulate in the nucleus and were used in fluorometric β-glucuronidase (GUS) assays. Images show merged fluorescent (eGFP: green) and bright field (BF) channels. Scale bars: 10 μm. (**B**) Schematic representation of the cell division-enabled leaf system (CDELS) on the scale of epidermal pavement cells in *N. benthamiana*. Differentiated cells in mature leaves (left panel) re-enter the canonical cell cycle upon ectopic expression of the *Arabidopsis* D-type cyclin CYCD3;1 (right panel). Nearly all re-activated pavement cells (diffuse yellow coloring) have completed cytokinesis after 3 days (filled arrowheads). See also *Figure 8—figure supplement 1B*. Scale bars: 20 μm. (**C**) Left panel: schematic representation of the CDEL system on the scale of the canonical cell cycle. Ectopically expressed CYCD3;1 targets and activates cyclin-dependent kinases (CDK, not shown), fostering the G1/S transition. Progression throughout the different cell-cycle phases is accompanied by the sequential activation of DNA synthesis (S) and mitotic genes (M). Histone *H4* and *KNOLLE* promoter-reporters were selected as readouts for G1/S and late G2/M transcriptional waves, respectively. Right panel: fluorometric GUS assay in tobacco leaf cells expressing the Medicago pKNOLLE::GUS (*pKNOLLE*) reporter construct in the absence or presence of ectopically expressed CYCD3;1. Fluorescence curves (GUS-mediated hydrolysis of 4 MUG) over time are shown for four biological replicates. Error bars indicate standard deviation. (**D**) Activity of GUS driven by the histone *H4* promoter (*pH4*) in the absence or presence of ectopically-expressed eGFP-NF-YA1 (NF-YA1) in CDEL samples (+CYCD3;1). All data points are shown, and crosses indicate sample means. Differences were not statistically significant (p-value = 0.3323) according to an unpaired t test with Welch's correction. Data are from three independent transformation experiments. n=12 ([-] NF-YA1) and 12 ([+] NF-YA1) biological samples. (**E**) Activity of GUS driven by the *KNOLLE* promoter (*pKNOLLE*) in CDEL samples (+CYCD3;1) in the absence or presence of ectopically-expressed eGFP-NF-YA1 variants with WT (NF-YA1) or mutated DNA-binding domain (NF-YA1[mutDBD]). All data points are shown, and crosses indicate sample means. Statistically significant differences ([-] NF-YA1 versus [+] NF-YA1; p-value <0.0001) or non-significant differences ([-] NF-YA1 versus [+] NF-YA1[mutDBD]; p-value = 0.7861) are based on a Kruskal-Wallis test followed by Dunn's multiple comparisons test. Data are from four to five independent transformation experiments. n=18 ([-] NF-YA1), 16 ([+] NF-YA1), 16 ([+] NF-YA1[mutDBD]) biological samples.

The online version of this article includes the following source data and figure supplement(s) for figure 8:

**Source data 1.** GUS activity (*pH4*) and statistical analysis.

*Figure 8 continued on next page*

*Figure 8 continued*

**Source data 2.** GUS activity (*pKNOLLE*) and statistical analysis.

**Figure supplement 1.** NF-YA1 modulates the entry into mitosis in a de-differentiating leaf cell population.

**Figure supplement 1—source data 1.** PDF file containing original immunoblots for *Figure 8—figure supplement 1A*, indicating the relevant bands.

**Figure supplement 1—source data 2.** Original files for immunoblot analyses displayed in *Figure 8—figure supplement 1A*.

**Figure supplement 1—source data 3.** Percentages of KNOLLE-positive cells and statistical analysis.

**Figure supplement 1—source data 4.** Percentages of split cells and statistical analysis.

**Figure supplement 2.** Constitutive expression of Medicago NF-YA1, but not of a MYB3R repressor, markedly competes with *KNOLLE* promoter activation in a CDEL system.

**Figure supplement 2—source data 1.** GUS activity and statistical analysis.

respective promoter sequences were previously employed to successfully drive the production of fusion proteins in roots of WT Medicago composite plants (*Su et al., 2023*; *Lace et al., 2023*). Weakly active in the pericycle and the vascular cylinder of non-inoculated roots (*Figure 9—figure supplement 1A*), the transcription of the two reporter constructs (pNPL::NLS:3xVenus / pENOD11::NLS:2x-mCherry) was strongly increased upon colonization of the root cortex, with cells hosting infection threads showing the highest activity. Expression driven from the *NPL* and *ENOD11* promoters gradually decreased in cells neighboring and underlying the infection site (*Figure 9—figure supplement 1B*) but remained high along the trajectory of infected cortical cells (*Figure 9—figure supplement 1C*).

For the next set of experiments, we selected the *ENOD11* promoter sequence to locally increase NF-YA1 levels in WT roots *via* the symbiotically induced expression of a non-fluorescently tagged version (pENOD11::FLAG-NF-YA1 or hereafter, pENOD11::NF-YA1). Limited to the area close to infection, additional *NF-YA1* expression led to a modest but significant reduction in the number of nodule primordia formed (*Figure 9—figure supplement 2A*) without impacting nodule colonization, with 93% of the primordia being infected in both the control condition and upon ectopic *NF-YA1* expression (n = 334 and 216, respectively; *Figure 9—figure supplement 2B–C*). To assess the influence of *NF-YA1* overexpression at the cellular level, we coupled the pENOD11::NF-YA1 construct to the *KNOLLE* G2/M transcriptional reporter (pKNOLLE::D-box-3xVenus-NLS; *Figure 9—figure supplement 3A–B*). As previously observed (*Figure 5—figure supplement 1C*), this destabilized nuclear reporter functions as a robust readout of a cell's competence to be crossed by an infection thread (low reporter signal) or to commit to cell division (high reporter signal) in Medicago roots. In this context, we sought to determine whether more cells become competent to host infection threads and whether NF-YA1 also regulates cell division activity when accumulated in the outer cortex.

A significant nuclear enlargement was observed in crossed cortical cells of both control roots and those additionally expressing *NF-YA1* (*Figure 9—figure supplement 4A–B*), implying that elevated levels of NF-YA1 did not prevent infected cells from proceeding through DNA replication. The local increase in *NF-YA1* expression also did not alter the typical expression profile of the transcriptional *KNOLLE* reporter (*Figure 9—figure supplement 4C*). Although overall reduced compared to control roots, the nuclear levels of triple-Venus fluorescence were kept low in infected cells and gradually increased in the vicinity of an infection site and within nodule primordia (*Figure 9—figure supplement 4D*) in transgenic roots ectopically expressing *NF-YA1* (*Figure 9—figure supplement 3A–B*). Competence for transcellular infection, defined here as a pre-mitotic arrest of the cell cycle, therefore appears to be maintained in the context of local *NF-YA1* overexpression. The latter leads to a slight (non-significant) increase in the number of infected cells per cortical layer: in almost 9% of the examined layers (C1 to C3, n=70), at least three cells hosted an infection thread upon ectopic *NF-YA1* expression, while this was observed only once in C3 (i.e. 3% of the examined cortical layers; n = 32) in control roots (*Figure 9—figure supplement 4E–G*). Regarding cell division activity, local overexpression of *NF-YA1* did not influence the proportion of adjacent cells dividing in the vicinity of infected cells (about 33% in each of the transgenic backgrounds; *Figure 9—figure supplement 5A–B*). By contrast, we observed a greater proportion of split cells on the trajectory of the cortical infection thread: 35% of infected cells (n=40) divided anticlinally in the outer cortex when *NF-YA1* expression was locally increased compared to only 12.5% (n=16) in control condition (*Figure 9A–B*). This supports the idea that, when accumulated in a symbiotic context, NF-YA1 may facilitate the division

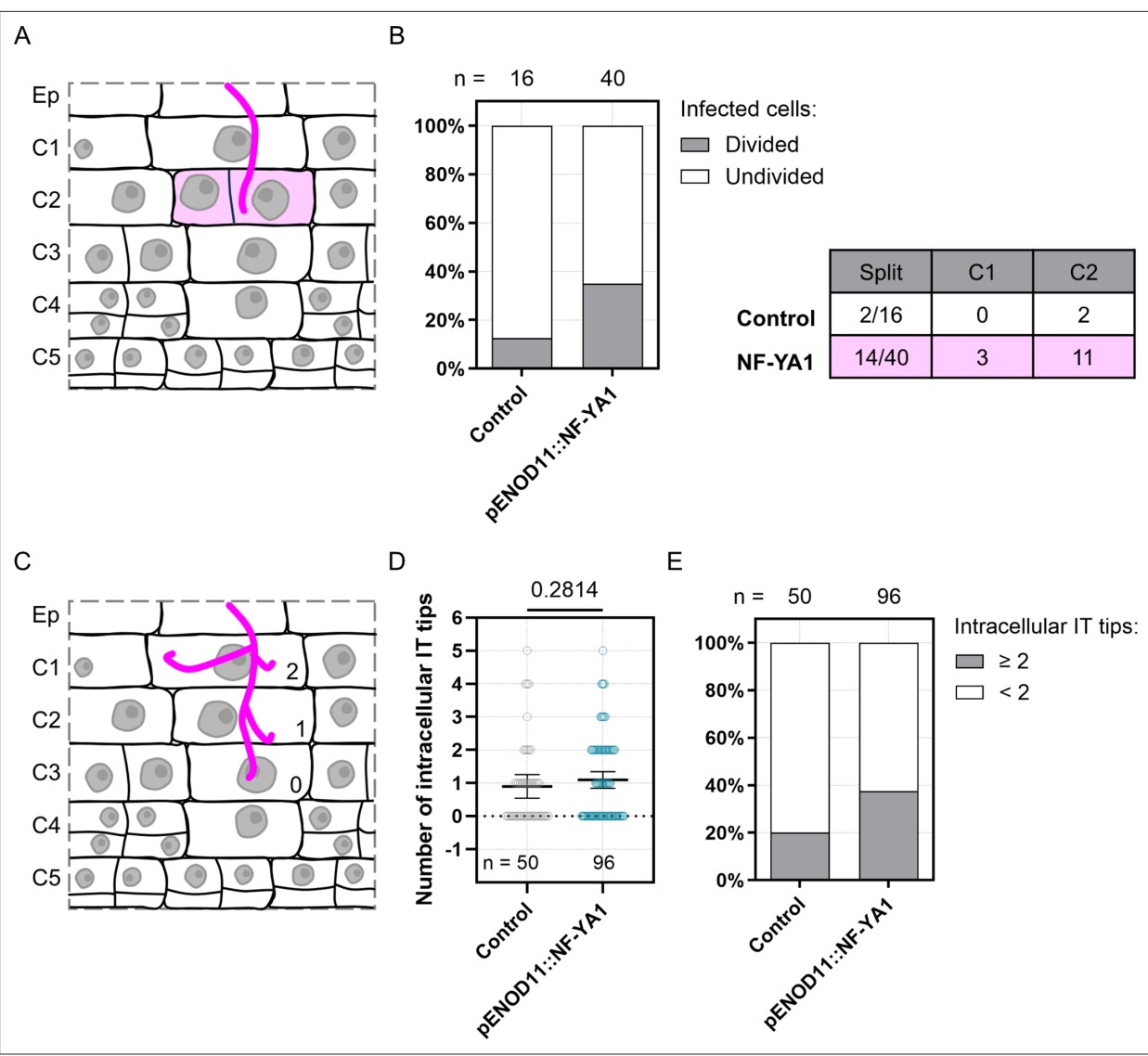

**Figure 9.** Ectopic expression of *NF-YA1* influences both cortical cell division on the trajectory of rhizobial infection and infection thread structure in WT transgenic roots. (**A**–**E**) Data were generated in Medicago WT roots expressing the *KNOLLE* G2/M transcriptional readout (pKNOLLE::D-box-3xVenus-NLS), without (Control) or with additional production of NF-YA1 on the cortical infection thread trajectory (pENOD11::NF-YA1), 3–6 dpi with *S. meliloti*. Two independent transformation experiments were performed. (**A**) Schematic representation illustrating the anticlinal division of an outer cortical cell (C2) positioned on the infection thread trajectory (diffuse magenta coloring). Rhizobia inside the infection thread are depicted in magenta. (**B**) Left panel: quantification of infected cells in outer cortical layers (C1 to C2) of early nodule primordia (stage I to III), divided anticlinally (gray) or undivided (white) in control transgenic roots or roots ectopically expressing Medicago *NF-YA1* (pENOD11::NF-YA1). The number (n) of cells analyzed in each transgenic background is indicated on the top. Right panel: table recapitulating the number and the cortical location of divided (split) infected cells in each transgenic background. Data are from 5 to 11 composite plants, with 9 (Control) to 21 (pENOD11::NF-YA1) nodule primordia analyzed. (**C**) Schematic representation illustrating the general structure of a cortical infection thread, with 2 (C1), 1 (C2) or no additional branch (0; C3) outside the main infection thread. Rhizobia are depicted in magenta. (**D**) Quantification of the number of infection thread (IT) tips outside the main IT, within individual cortical cells being passed or already fully crossed. All data points are shown, and horizontal bars indicate sample means with a 95% confidence interval. Differences were not statistically significant (p-value = 0.2814) according to a Mann-Whitney test. (**E**) Quantification of infected cortical cells in control transgenic roots or roots ectopically expressing *NF-YA1*, showing 2 or more (gray) or only 1 (white) additional IT tip outside the main IT. (**D**–**E**) n=50 (Control) or 96 (pENOD11::NF-YA1) infected cortical cells (C1 to C4). Data are from 8 to 14 composite plants, with 14 (Control) to 28 (pENOD11::NF-YA1) nodule primordia analyzed.

The online version of this article includes the following source data and figure supplement(s) for figure 9:

**Source data 1.** Number of intracellular IT tips and statistical analysis.

**Figure supplement 1.** Infection-induced *ENOD11* and *NPL* transcriptional reporters are activated in the direct vicinity of cortical infection threads.

*Figure 9 continued on next page*

*Figure 9 continued*

**Figure supplement 2.** Overexpression of *NF-YA1* on the cortical infection thread trajectory has no major impact on nodule primordium colonization.

**Figure supplement 2—source data 1.** Number of nodule primordia per transformed root and statistical analysis.

**Figure supplement 3.** Overview of transgenic Medicago roots expressing the *KNOLLE* G2/M transcriptional readout in the absence or presence of ectopically expressed *NF-YA1*.

**Figure supplement 4.** A slightly larger proportion of cells is competent for infection upon local increase in *NF-YA1* expression.

**Figure supplement 4—source data 1.** Nuclear area and statistical analyses.

**Figure supplement 4—source data 2.** Corrected total *pKNOLLE* nuclear fluorescence and statistical analyses.

**Figure supplement 4—source data 3.** Number of infected cells per cortical layer and statistical analysis.

**Figure supplement 5.** The division of cells adjacent to the infection thread is not affected by ectopic expression of *NF-YA1*.

of endoreduplicated cortical cells in anticipation of infection. Conversely, increased levels of NF-YA1 presumably repress the transition to mitosis within cells undergoing infection (*Figure 9—figure supplement 4D*) and appear to also influence the general structure of infection threads (*Figure 9C–E*, *Figure 9—figure supplement 3B*). Ectopic expression of *NF-YA1* led to a higher proportion of cells forming supernumerary infection thread branches, with 37.5% of infected cells showing at least two additional branches outside the main infection thread (n=96) compared to 20% in control condition (n=50; *Figure 9E*). Accumulated on the trajectory of a cortical infection, NF-YA1 might therefore act in increasing the proportion of potential host cells while promoting the branching of infectious structures in the Medicago root cortex.

## Discussion

Failure to timely coordinate infection and organogenesis programs in the root cortex results in the formation of poorly colonized nodules that are unable to support the plant's demand to obtain fixed nitrogen (*Guan et al., 2013*; *Laporte et al., 2014*; *Lee et al., 2024*). Here, we sought to understand at cellular resolution whether cell-cycle progression and cellular reprogramming are altered in a spatially confined array of host cells that gain a time-limited stage of competence for transcellular infection.

Our data support a model where dividing cells contributing to nodule formation, including target cells for cortical infection, are polyploid and contain 4 copies (C) of each chromosome (*Figure 6D–E*). In Medicago, these 4C cells either pre-exist in the root cortex (*Vinardell et al., 2003*; *Carotenuto et al., 2019a*; *Carotenuto et al., 2019b*) or arise from a single endoreduplication round prior to their possible entry into cell division (*Libbenga and Torrey, 1973*). As competent cells are crossed by a cortical infection thread or fully internalize bacteria, and despite being embedded into a proliferating tissue, they reach an 8C DNA content (*Figures 7C and 10*) but without committing to a subsequent cell division (*Figures 4B–D and 5A–B*). Rather, cells supporting transcellular infections presumably use a cell-cycle GAP phase after DNA replication (*i.e.,* G2) and extend such phase to remodel their histone H3 composition (*Figure 2A–B and D*). Finally, this differentiation process and concomitant transcellular passage are accomplished within a single, last cell-cycle round (*Figure 1A*; *Otero et al., 2016*). Here, and similar to infected root hairs (*Breakspear et al., 2014*), a rapid switch to another endocycle is likely prevented as both G1- and S-phase markers are kept low (*Figure 4D*, *Figure 5—figure supplement 2C*). Taken together, these events control a cell-cycle progression state where infection players operate to ensure a successful transmission of rhizobia to the first nodule cells.

### Modulating endopolyploidy during rhizobial infections

In line with findings from *Arabidopsis* reporting that a downregulation of *H3.1* gene expression and a massive eviction process contribute to various differentiation programs, for example, in the male and female gametes (*Ingouff et al., 2010*), the root meristem, the embryo (*Otero et al., 2016*) and the female germline precursor (*Hernandez-Lagana and Autran, 2020*), we show that the genetic program enabling intracellular infections of Medicago roots accesses the same mechanism (*Figure 2A–B*, *Figure 2—figure supplement 1C–E*). Reciprocally, our data indicate that high H3.1 levels tend to be maintained in infected cells with blocked or aborted infections (*Figure 3A,C*, *Figure 3—figure supplements 1 and 2*). Furthermore, the nuclear enlargement in infected cells (*Figure 2C*) indicates profound transcriptional reprogramming and activation of new gene regulatory networks, as

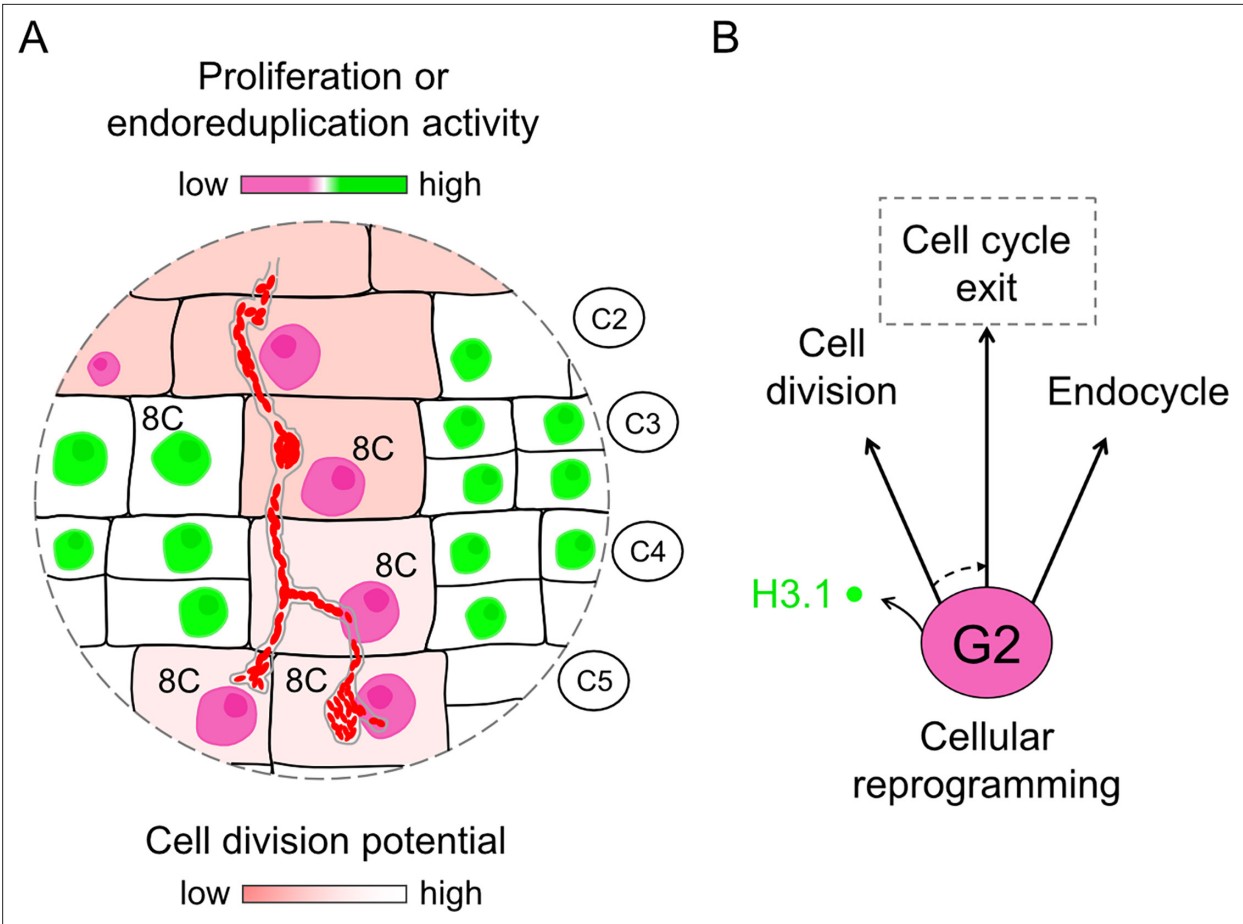

**Figure 10.** Model illustrating characteristic cellular traits on the cortical infection thread trajectory. (**A**) Cortical cells competent for sustained transcellular infection thread progression are characterized by a reduced proliferation or endoreduplication activity (pink to green two-color gradient). By the time they are crossed and ultimately internalize bacteria, such tetraploid cells reach a post-replicative, 8C DNA content. But, contrary to direct neighbor cells, they do not commit to a subsequent cell division (red to white two-color gradient). (**B**) Rather, cells supporting transcellular infection may enter a prolonged G2-phase during which they remodel their histone H3 composition as part of their cellular reprogramming for infection. This differentiation process and concurrent transcellular passage are accomplished within a last cell-cycle round. We propose that cellular factors controlling the G2/M transition and eventually the exit from the canonical cell cycle (dashed arrow) are prominent candidates to support intracellular infection competence.

previously shown to occur in LCO-treated (*Knaack et al., 2022*) or young symbiotic cells (i.e. infected root hairs and 8C cells from mature nodules; *Breakspear et al., 2014*; *Nagymihály et al., 2017*; *Liu et al., 2019a*). We expanded the latter view by showing that tetraploid cortical cells passed by an infection thread also reach a presumable 8C DNA content (*Figure 7C*). As reported for growing cell types in Arabidopsis (*Bhosale et al., 2018*), such alteration in endopolyploidy might support, among others, changes in cell wall composition, which have also recently been demonstrated to occur specifically around infected cells in Medicago (*Su et al., 2023*; *Gaudioso-Pedraza et al., 2019*). In the determinate nodulator *L. japonicus*, polyploidization of early nodule cells might be controlled by *vagrant infection thread 1* (*VAG1*), a putative component of the DNA topoisomerase VI (topo VI) complex, as cortical cells with reduced ploidy levels cannot be infected in the *vag1* mutant (*Suzaki et al., 2014*).

Interestingly, but different from infections by mycorrhizal fungi where repetitive rounds of endoreduplication occur (*Russo and Genre, 2021*) and support fungal colonization (*Figure 1C*; *Carotenuto et al., 2019a*; *Carotenuto et al., 2019b*), markedly reduced histone H3.1 contents in cells maintaining rhizobial infections in the root cortex up to the first recipient cells in nodule primordia indicate reduced endocycling rates (*Figures 1A and 2D*; *Otero et al., 2016*). Such attenuation of cell-cycle progression, together with decreased transcript levels of mitotic cyclins, was shown to occur in budding yeast and *Arabidopsis* (*Negishi et al., 2016*; *Gigli-Bisceglia et al., 2018*) following the activation of a

cell-wall integrity (CWI) checkpoint. More recently, Ma and colleagues proposed a scenario where the CWI kinase THESEUS1 (THE1), with a related gene being induced in infected root hairs (*Breakspear et al., 2014*; *Liu et al., 2019a*), triggers cell wall stiffening as a response to endoreduplication attenuation (*Ma et al., 2022*). A similar feedback mechanism, if involved, might enable to adjust cell wall modifications associated with infection thread growth and transcellular passage.

## A G2-phase arrest and attenuated endoreduplication as a plausible context for cortical infections

Reaching and staying in a pre-mitotic stage has been proposed to facilitate the long-distance progression of infection threads in legume species forming so-called cortical, pre-infection threads (PITs; *Yang et al., 1994*; *Timmers et al., 1999*; *Foucher and Kondorosi, 2000*). These microtubule-rich cytoplasmic bridges resemble the onset of a cell division plane that requires the formation of a cortical division zone, which is established from late S- through G2 phase (*Costa, 2017*). Interestingly, this cell-cycle progression window coincides with the one we observed in infected cortical cells (*Figures 1A and 2A–B* and *Figure 4A–D*), reinforcing the hypothesis that part of the machinery activated in anticipation of cytokinesis could be recruited and assist transcellular infection of PIT-forming cells (*Murray, 2016*). This is in line with a recent report showing that a rhizobium-induced mitotic module, comprising an α-Aurora kinase gene and its MYB3R transcriptional activator, supports endosymbiotic infection in Medicago (*Gao et al., 2022*). Interestingly though, overexpression of the activator-type *MYB3R1* gene resulted in the formation of abnormally bulbous or branched infection threads (*Gao et al., 2022*). These data emphasize the need to balance the activity of transcriptional regulators driving mitosis and cytokinesis in infected cells.

In summary, we hypothesize that endocycle suppression after a round of DNA replication (*Breakspear et al., 2014*) and exiting the cell cycle from the G2 phase might therefore create a permissive, yet controlled state for cortical infection thread progression, where polarization and cell-wall remodeling factors specialized for intracellular infection (*Lace et al., 2023*; *Su et al., 2023*) act in a defined cell-cycle window to control infection thread growth and guidance across cortical layers. Although differences in the relationship between rhizobial infection and endoreduplication (*Fan et al., 2022*) or cell division (*Monroy-Morales et al., 2022*) exist in legumes forming determinate or indeterminate nodules, it is important to note that a post-replicative state of the cell cycle remains most favorable for infection.

## Endopolyploidization following cytokinin perception: a legume innovation?

Using the centromeric histone H3 (CENH3) fluorescent reporter enabled us to assess the number of chromosomes, the progression through the G2 phase (*Figure 7—figure supplement 1A–B*; *Lermontova et al., 2006*), and to clearly identify diploid and polyploid cells (*Figure 6B and E*; *De Storme et al., 2016*). Combining CENH3 patterns with the histone H3.3 marker showed that somatic cortical cells giving rise to Medicago root nodules are mainly tetraploid, which is consistent with previous observations in legumes producing indeterminate nodules (*Wipf and Cooper, 1938*; *Torrey and Barrios, 1969*). In addition, our imaging data indicate that dividing endopolyploid cells, despite an elevated risk of genome instability, are homogenously maintained in the nodule tissue (*Figure 6—figure supplement 2B–C*). This suggests that tetraploidization is an important feature for nodule fitness and infectability, as it may enable processes such as the required integration of many contextual stimuli (*e.g.*, bacterial signals; *Russo and Genre, 2021*). Furthermore, our findings point towards tetraploidization of cortical cells being a hallmark for nodule identity, as pericycle cells in nodule or lateral root primordia remain strictly diploid (*Figure 6E*, *Figure 6—figure supplement 2C*). Interestingly, when dissected and cultured in suspension, cortical cells from the susceptible zone of pea roots entered cell divisions as tetraploid cells only when cytokinin (kinetin) was applied in addition to auxin (2,4-D; *Libbenga and Torrey, 1973*). This is particularly interesting since a specific cortical response to cytokinin, an auxin/cytokinin interplay, and subsequent cellular proliferation in the root cortex appear to be a legume innovation (*Guan et al., 2013*; *Gauthier-Coles et al., 2019*). Whether an early somatic, whole-genome duplication event is a signature of functional root nodules and part of this innovation, as recently suggested for the recruitment of meristematic regulators (*Dong et al., 2021*), remains to be investigated.

## NF-YA1, possible intermediary between intracellular infection and cell-cycle control

While this and other reports have shed light onto cell-cycle patterns during rhizobial infections in roots and nodules, genetic factors coordinating cell-cycle progression and intracellular infections remained enigmatic. Using the heterologous CDEL system (*Xu et al., 2020*) has enabled us to test the impact of at least one of the potential candidates, the symbiotic transcription factor NF-YA1, on cell-cycle-dependent activities in a de-differentiating cellular population. Expression of NF-YA1 in this system did not prevent cells from passing through S-phase (*Figure 8D*). But, while the activated cells (as indicated by the induction of the KNOLLE reporter) still engage in mitosis and eventually divide (*Figure 8—figure supplement 1B*), significantly fewer transformed cells promoted the expression of this G2/M marker at our experimental timepoints (*Figure 8E*, *Figure 8—figure supplement 1C*). These results strongly suggest that, in the CDEL system, NF-YA1 negatively regulates the entry into mitosis despite the presence of CYCD3;1. The latter has been shown to promote mitotic cycles and to restrain differentiation and endoreduplication when accumulated in shoot tissues (*Schnittger et al., 2002*; *Dewitte et al., 2003*). Our observations are therefore consistent with the indispensable role of A-type NF-Y subunits in the transition to the early differentiation and specification of nodule cells, downstream of cytokinin perception and initial cell-cycle reactivation in Lotus and Medicago. According to loss- and gain-of-function studies, adequate levels of NF-YA1 appear indeed essential to generate a pool of enlarged (likely endoreduplicated) cells, deriving from a functional meristem in Medicago nodules and competent to host nitrogen-fixing bacteria (*Combier et al., 2006*; *Xiao et al., 2014*; *Hossain et al., 2016*; *Schiessl et al., 2019*; *Lee et al., 2024*). Interestingly, *CYCD3;1* and other genes positively regulating the progression of cells through G2 and mitosis are directly controlled by a key regulator of nodule identity, light-sensitive short hypocotyl1 (LSH1; *Lee et al., 2024*). It was proposed that LSH1/LSH2 transcription factors foster the proliferation of cortical cells that support bacterial colonization in part *via* promoting *NF-YA1* expression in the newly divided cells associated with the specification of the nodule primordium. At an earlier stage of infection, forcing the expression of Medicago *NF-YA1* in the outer cortex revealed its dual role in promoting cell division and intracellular infection competence on the trajectory to the nodule cells (*Figure 9A–E*, *Figure 9—figure supplement 4A–G*). Although the effects we observed are subtle, they might be explained by the ability of NF-YA1 to modulate the G2/M transition. A couple of recent studies in *Arabidopsis* demonstrate that meristematic regulators from the GRAS and AP2 transcription factor families drive cell proliferation while enabling the local accumulation of CDK inhibitors, thereby controlling the duration of G1- and G2 phases (*Nomoto et al., 2022*; *Echevarria et al., 2022* – pre-print). Whether LSHs and/or NF-YA subunits work similarly, and whether genes responsible for structuring the infection thread are part of the NF-YA1 regulon are exciting possibilities that remain to be explored.

In summary, we propose that local control of cell-cycle activity and a cell-cycle exit decision along the infection thread trajectory provide a window of cellular competence for cortical infections to progress down to the nodule primordium niche in Medicago. We demonstrated that the cell division and differentiation potentials are controlled in a spatially confined cell file in the root cortex. We also showed that post-replicative, tetraploid cells reaching an 8C DNA content are optimal for intracellular infection. Consequently, it is tempting to hypothesize that exactly these cells comprise a highly specific transcriptional and regulatory landscape that ultimately defines the molecular susceptibility state for intracellular infection competence.

## Materials and methods

### Key resources table

| Reagent type (species) or resource | Designation | Source or reference | Identifiers | Additional information |
|---|---|---|---|---|
| Gene (*Medicago truncatula*) | H3.1 | *M. truncatula* A17 r5.0 genome portal | MtrunA17_Chr8g0383781 | v5.1.9 |
| Gene (*M. truncatula*) | H3.3 | *M. truncatula* A17 r5.0 genome portal | MtrunA17_Chr4g0054161 | v5.1.9 |
| Gene (*M. truncatula*) | CENH3 | *M. truncatula* A17 r5.0 genome portal | MtrunA17_Chr8g0347231 | v5.1.9 |

*Continued on next page*

*Continued*

| Reagent type (species) or resource | Designation | Source or reference | Identifiers | Additional information |
|---|---|---|---|---|
| Gene (*M. truncatula*) | *KNOLLE* | *M. truncatula* A17 r5.0 genome portal | MtrunA17_Chr5g0398891 | v5.1.9 |
| Strain, strain background (*M. truncatula* cultivar Jemalong A17) | Medicago; WT | Heritage Seeds Pty. Ltd., Adelaide, AU | Jemalong | |
| Strain, strain background (*Agrobacterium rhizogenes*) | ARqual | other | | Lab stocks |
| Strain, strain background (*Rhizophagus irregularis*) | *R. irregularis* | Agronutrition | Agronutrition:DAOM 197198 | |
| Strain, strain background (*Escherichia coli*) | Stbl3 | Invitrogen, Thermo Fisher Scientific | Thermo Fisher Scientific:C737303 | |
| Genetic reagent (*M. truncatula* ecotype A17 FNB-induced deletion mutant) | *daphne-like* | *Liu et al., 2019b*; PMID:30610167 | FN8113 | Rene Geurts |
| Genetic reagent (*M. truncatula* ecotype A17 EMS mutant) | *nf-ya1-1* | *Laporte et al., 2014*; PMID:24319255 | *nf-ya1-1* | |
| Genetic reagent (*Sinorhizobium meliloti* strain 2011 *mCherry*) | *S. meliloti* mCherry | other | | Lab stocks |
| Genetic reagent (*S. meliloti* strain 2011 *GFP*) | *S. meliloti* GFP | other | | Lab stocks |
| Genetic reagent (*S. meliloti* strain 2011 *pXLGD4 lacZ*) | *S. meliloti* lacZ | other | | Lab stocks |
| Genetic reagent (*Agrobacterium tumefaciens* strain GV3101 p19) | p19 silencing suppressor | *Silhavy et al., 2002*; PMID:12065420 | | |
| Antibody | anti-GFP (Mouse monoclonal) | Takara Bio | Takara:632381; RRID:AB_2313808 | WB (1/5000) |
| Antibody | anti-DsRed (Rabbit polyclonal) | Takara Bio | Takara:632496; RRID:AB_10013483 | WB (1/5000) |
| Antibody | anti-HA-Peroxidase (Rat monoclonal) | Roche | Roche:12013819001; RRID:AB_390917 | WB (1/2000) |
| Antibody | goat anti-mouse (Goat polyclonal, HRP conjugate) | Sigma-Aldrich | Sigma-Aldrich: A4416; RRID:AB_258167 | WB (1/2000) |
| Antibody | goat anti-rabbit (Rabbit polyclonal, HRP conjugate) | Sigma-Aldrich | Sigma-Aldrich: A6154; RRID:AB_258284 | WB (1/2000) |
| Recombinant DNA reagent | Golden Gate L0 (plasmids) | GeneArt, Thermo Fisher Scientific | | Distributed via https://www.ensa.ac.uk/ |
| Recombinant DNA reagent | Golden Gate L1 and L2 (plasmids) | This paper | | See - *Supplementary file 1* |
| Recombinant DNA reagent | PlaCCI (plasmid) | *Desvoyes et al., 2020*; PMID:32989288 | | Bénédicte Desvoyes; Crisanto Gutierrez |
| Chemical compound, drug | Paraformaldehyde (PFA) | Sigma-Aldrich | Sigma-Aldrich:158127; CAS:30525-89-4 | |
| Chemical compound, drug | Glutaraldehyde 25% | Carl ROTH | Carl ROTH:4157.3 | |
| Chemical compound, drug | Plaque Agarose; low melting temperature agarose | Biozym | Biozym:840101 | |
| Chemical compound, drug | Fluorescent Brightener 28; Calcofluor White | MP Biomedicals | MB Biomedicals:158067; CAS:4404-43-7 | (0.1% w/v) |
| Chemical compound, drug | 5-bromo-4-chloro-3-indolyl-β-D-galactopyranoside (X-β-Gal) | Carl ROTH | Carl ROTH:2315.3 | |
| Chemical compound, drug | 4-Methylumbelliferyl-β-D-glucuronide (4 MUG) | Carl ROTH | Carl ROTH:6394.2 | |
| Software, algorithm | Fiji/ImageJ | *Schindelin et al., 2012*; PMID:22743772 | Fiji (RRID:SCR_002285) | |
| Software, algorithm | FigureJ (ImageJ plugin) | *Mutterer and Zinck, 2013*; PMID:23906423 | | v1.36 |

*Continued on next page*

*Continued*

| Reagent type (species) or resource | Designation | Source or reference | Identifiers | Additional information |
|---|---|---|---|---|
| Software, algorithm | NIS-Elements D | Nikon GmbH | NIS-Elements (RRID:SCR_014329) | v5.20.00 |
| Software, algorithm | Jalview | *Procter et al., 2021*; PMID:33289895 | Jalview (RRID:SCR_006459) | v2.11.2.6 |
| Software, algorithm | GraphPad Prism | GraphPad Software Inc. | GraphPad Prism (RRID:SCR_002798) | v9.5.1 and v10.4.1 |
| Other | Fahraeus medium | *Boisson-Dernier et al., 2001*; PMID:11386364 | | |
| Other | BsaI-HFv2 (type IIS restriction enzyme) | New England BioLabs | New England BioLabs:R3733S | |
| Other | BpiI / BbsI-HF (type IIS restriction enzyme) | New England BioLabs | New England BioLabs:R3539S | |
| Other | T4 DNA ligase | New England BioLabs | New England BioLabs:M0202S | |
| Other | ClearSee (clearing solution) | *Ursache et al., 2018*; PMID:29171896 | | |

## Plant materials and growth conditions

The *Medicago truncatula* (Medicago) wild-type (WT) ecotype A17, cultivar Jemalong was used in this study. The *daphne-like* (FN8113) and *nf-ya1-1* mutant lines have been described previously (*Liu et al., 2019b*; *Laporte et al., 2014*). Seeds from the *daphne-like* mutant were kindly provided by Rene Geurts (Laboratory of Molecular Biology, Wageningen University, The Netherlands).

Seeds were scarified by covering them with sulfuric acid ($H_2SO_4$) 96% for 8 min and washed six times with sterile tap water. After surface sterilization with a bleach solution (1.2% sodium hypochlorite, NaClO and 0.1% sodium dodecyl sulfate, SDS) for 1 min, the six washing steps were repeated and seeds were sown on 1% (w/v) agar plates. Seeds were then stratified at 4 °C for 5 days in the dark before being germinated at 22 °C in the dark for 16 hr.

## Hairy root transformation

Composite Medicago plants were generated with transgenic hairy roots as previously described (*Boisson-Dernier et al., 2001*). *Agrobacterium rhizogenes* ARqual cells carrying the desired binary vector were cultured in liquid LB medium containing appropriate antibiotics at 28 °C, shaking overnight. Three hundred µl of the culture (OD 600 nm: 0.5–0.7) were plated on solid LB medium with corresponding antibiotics and incubated at 28 °C for 2 days. Plants were prepared by removing the seed coat from germinated seedlings, and about 5 mm of the root tip region was removed before the cut site was dipped into the solid *A. rhizogenes* culture. Up to 25 inoculated seedlings were placed on solid Fahraeus medium supplemented with 0.5 mM $NH_4NO_3$. Transformed seedlings were then grown vertically in a climate chamber at 21 °C for 3 days in the dark. After another 4 days at 21 °C (16 h light/8 hr dark, 35 µmol m$^{-2}$ s$^{-1}$ light intensity), with only the roots protected from illumination, composite plants were transferred to fresh solid Fahraeus plates (0.5 mM $NH_4NO_3$) and grown for 10 days at 22 °C following the same photoperiod. Transgenic roots were visualized based on the fluorescence emitted by nuclear markers (*M.t.* pH3.3::H3.3-mCherry; *A.t.* or *M.t.* pH3.1::H3.1-mCherry; pLjUbi::NLS-2xmCherry) or betalain production (pSlEXT1::RUBY) using an AXIO Zoom.V16 stereomicroscope (Zeiss, Oberkochem, Germany). Un-transformed roots were removed, and the composite plants were transferred in pots (7x7 × 8 cm) containing a 1:1 (v/v) mixture of quartz sand (0.1–0.5 mm grain size; Sakret, Berlin, Germany) and vermiculite (0–3 mm; Ökohum, Herbertingen, Germany) equilibrated with 25 ml of a ¼ Hoagland solution (0.1 mM $KNO_3$). After 4 days of acclimation (16/8 hr light/dark photoperiod, 24 °C temperature, 60% humidity and 70–90 µmol m-2 s-1 light intensity), plants were subjected to rhizobial inoculation.

For this, *Sinorhizobium meliloti* (Sm2011) strains producing mCherry, GFP, or expressing the *lacZ* reporter were grown on solid TY medium containing 10 µg/ml tetracycline for 2 days at 28 °C. A single colony was used as an inoculum, and bacteria were grown in liquid TY medium containing the antibiotic at 28 °C, shaking for 2 days. Cells were centrifuged for 8 min at 2205 × *g*, washed once with liquid Fahraeus medium, and finally resuspended into the same medium to reach an OD 600 nm of 0.0015. Five ml of this bacterial suspension were applied to each composite plant. Pots were watered

twice a week with tap water and fertilized with a ¼ Hoagland solution (0.1 mM KNO$_3$) once a week. Transgenic roots were harvested at different time points (3–14 days) for tissue fixation, clearing, and deep tissue imaging by confocal microscopy, or stained for β-galactosidase activity when inoculated with the Sm2011-*lacZ* strain.

## Construct design

For expressing the canonical (*H3.1*) and replacement (*H3.3*) histone genes of Medicago, genomic fragments containing the promoters (*H3.1*: 2623 bp; *H3.1* (2): 2580 bp; *H3.3*: 3245 bp) and open reading frames (ORF) except the termination codon were used. The pH4::GUS reporter used in quantitative fluorometric assays comprised a 2080 bp fragment of the promoter. To apply the PlaCCI tool to legume-rhizobia interactions, the Medicago *CDT1a* genomic sequence was used and placed under control of a 2548 bp promoter fragment. Expression of the centromeric histone (*CENH3*) gene (coding sequence) was driven by a tentative promoter comprising 1000 bp upstream of the start codon. Transcriptional *KNOLLE* reporters were driven by a 2325 bp-long fragment of the *KNOLLE* promoter. The core pentamer of all three mitosis-specific activator (MSA) cis-elements (AACGG) within the *KNOLLE* promoter was mutated by changing two bases (ATTGG) according to *Ito et al., 2001*. The *KNOLLE* genomic sequence was used to build the translational reporter (pKNOLLE::eGFP-KNOLLE). For Medicago NF-YA1, the genomic sequence was used, and the NF-YA1$^{mutDBD}$ variant was created by replacing the conserved RGC amino acid residues (positions 213–215) by an AAA triplet according to *Laloum et al., 2014*. A 1003 bp-long fragment of the *ENOD11* promoter was used to build transcriptional reporters and to drive additional *NF-YA1* expression on the infection trajectory. The transcriptional *NPL* reporter was built using a 2038 bp-long promoter sequence. For the repressor-type *MYB3R* transcription factor, the coding sequence (CDS) was used. All Medicago sequence information was obtained from the Mt4.0v1 genome through Phytozome.

The *Arabidopsis thaliana* (*Arabidopsis*) pCYCB1;2::N-CYCB1;2-NLS-3xVenus reporter construct was designed according to *Trolet et al., 2019* and included a genomic fragment (AT5G06150) of 1929 bp, comprising 1090 bp of promoter sequence, the 5' untranslated region and 839 bp from the ORF including the first intron. Addition of two bases (GG for Gly) at the 3' end of this genomic fragment enabled the production of the N-terminal CYCB1;2 domain (including the destruction box [D-box] motif) fused in-frame to a triple-Venus nuclear reporter (*Binder et al., 2014*; *Nadzieja et al., 2019*) or to mCitrine in the adapted version of the PlaCCI sensor (PlaCCI v2). The CYCB1;1 N-terminal (D-box) domain was produced from a genomic fragment (AT4G37490) of 713 bp including the first 2 introns (nucleotides [nts] 41–308 and 488–584) and 129 bp of the third exon (nts 585–713). This module was fused in-frame to a triple-Venus nuclear fluorescent protein (FP) to generate a destabilized version of the fluorescent reporter. The CYCD3;1 (AT4G34160) CDS was placed under control of the cauliflower mosaic virus (CaMV) 35 S promoter (426 bp) and fused in frame to a 3xHA module to build the p35S::CYCD3;1 HA construct. *Arabidopsis* sequence data can be found on Phytozome (*Arabidopsis thaliana* TAIR10 annotation version). To produce the C-terminally truncated, hyperactive version of the *Nicotiana tabacum* MybA2 transcriptional activator (pLjUbi::eGFP-NtMybA2Δ630), the CDS (GenBank accession: AB056123.1) coding for the first 630 amino acid residues was used (*Araki et al., 2004*). The *RUBY* reporter (*He et al., 2020*) was placed under control of the promoter from the *Solanum lycopersicum extensin 1* gene (pSlEXT1), showing epidermal expression in Medicago (*Su et al., 2023*) and was used as a selection marker for transformed roots. The original GoldenBraid PlaCCI transformation plasmid was kindly provided by Bénédicte Desvoyes and Crisanto Gutierrez (Centro de Biologia Molecular Severo Ochoa, CSIC-UAM, Madrid, Spain).

## Construct production using the Golden Gate cloning system

The Golden Gate modular cloning (MoClo) system was used to generate binary plasmids for transgene expression in plant cells (*Weber et al., 2011*). Promoter, genomic and coding sequences were synthesized and cloned into pMS-RQ (GeneArt, Thermo Fisher Scientific, Regensburg, Germany). Level 0 (L0) functional modules used in this study are held for distribution in the ENSA project collection (https://www.ensa.ac.uk/). Details of the L1 expression cassettes and L2 binary vector composition, including used promoters, genetically encoded FPs (stable or short-lived) or epitope tags and the accession numbers for requesting constructs, are provided in a supplementary file. One-pot reactions using the Type IIS restriction enzymes BsaI-HFv2 (R3733S) and BpiI / BbsI-HF (R3539S) and the T4

DNA ligase (M0202S), all from New England BioLabs (Frankfurt am Main, Germany), were assembled according to *Binder et al., 2014*. Cut-ligation reactions were cycled 50 times between 37 °C for 2 min and 25 °C for 5 min, followed by 5 min at 50 °C and 5 min at 80 °C. Five µl of the reaction were transformed into *Escherichia coli* Stbl3 (for L1 and L2 plasmids carrying histone H3.1 or H3.3 expression cassettes) or TOP10.

## Root colonization by *Rhizophagus irregularis*

Following excision of untransformed roots, composite Medicago plants were transferred to chive (*Allium schoenoprasum*) nurse pot (9x9 cm) systems containing *R. irregularis* (DAOM 197198, sterile spore inoculum from Agronutrition, Carbonne, France) set up according to *Demchenko et al., 2004* (4 plants per pot). Each pot was covered with a transparent plastic bag (20x30 cm) for two days to keep plants in moisturized conditions. Plants were grown in a controlled environment chamber at 24 °C with a 16/8 hr light/dark photoperiod, with each pot being watered every second or third day with 50 ml deionized water and fertilized once per week with 50 ml ¼ Hoagland medium (32 µM $KH_2PO_4$). After 7–15 days of co-culturing, composite plants were harvested, and excised transgenic roots were subjected to fixation, clearing, and Calcofluor White staining prior to imaging.

## Clearing and staining of the root material

The presence of all expected fluorescent reporters was verified in the meristematic region using an AXIO Zoom.V16 stereomicroscope (Zeiss) before further analyzing the transgenic roots. Fixation, clearing, and cell-wall counter-staining procedures were performed according to *Ursache et al., 2018*. Briefly, transformed roots were isolated and fixed in a freshly prepared 4% paraformaldehyde (PFA) solution in 1 x phosphate-buffered saline (PBS), infiltrated under vacuum for 15 min and then gently agitated for 60–120 min at room temperature (RT) in the dark. After two washes in 1 x PBS, roots were transferred to the clearing (ClearSee) solution and incubated 7–10 days with gentle shaking, changing the ClearSee solution every 2 days. Prior to imaging, roots were stained with 0.1% Calcofluor White (MP Biomedicals, Eschwege, Germany) in ClearSee for 1–2 hr, before being rinsed once and washed in the clearing solution for 30 min. Calcofluor White was imaged using a 405 nm laser diode excitation source and a 425–475 nm detection window.

## Nodule sections

To assess the localization pattern of H3.1 and H3.3 histone variants across different nodule zones, composite plants were generated by hairy root transformation as described above. Nodules were harvested 35–40 days after inoculation in open pots and fixed in 4% (w/v) PFA in 1 x PBS, under vacuum treatment for 15 min and then gently agitated for 45 min at RT. The fixed samples were embedded in 6% (w/v) low melting temperature agarose (Biozym Scientific, Hessisch Oldendorf, Germany), and longitudinal sections (70 µm thickness) were obtained using a VT1000S vibratome microtome (Leica Biosystems, Mannheim, Germany).

## Fluorescent reporter imaging by Confocal Laser-Scanning Microscopy (CLSM)

Imaging of plant material (i.e. nodule semi-thin sections and cleared root samples) was performed using a TCS SP8 confocal microscope (Leica Microsystems) controlled by the LAS X v3 software. Images were acquired with a 20 x/0.75 or a 40 x/1.10 (HC PL APO CS2) water immersion objective. Genetically encoded FPs were excited with an argon laser or a White Light Laser (WLL) source. Emitted photons were collected using hybrid detectors (HyDs). eCFP and mTurquoise2 were excited with an argon laser line at 458 nm and the emission was detected at 470–510 nm. All other FPs were excited with WLL excitation (ex) lines, and emissions (em) were detected according to the following settings: eGFP = 488 nm (ex) / 500–550 nm (em); YFP = 515 nm (ex) / 525–586 nm (em); mCitrine and Venus = 515 nm (ex) / 525–550 or 525–570 nm (em); mCherry: 561 or 587 nm (ex) / 600–650 or 620–650 nm (em).

## Image analysis and quantification of nuclear fluorescent signals

All images were processed and analyzed using the Fiji/ImageJ open-source software (*Schindelin et al., 2012*). Image files were used to create figure panels through the FigureJ plugin (*Mutterer and*

Zinck, 2013). If not indicated differently, 2D maximum intensity projections of z-stacks are presented

in the figures. In the Green Fire Blue color scheme, blue or yellow indicate low or high fluorescence levels, respectively. All quantitative measurements were performed on single focal planes and single channels from unprocessed z-stacks, acquired from fully transformed Medicago roots. The Fire lookup table (LUT) was applied to facilitate the visualization of low-intensity pixels. Nuclear areas were measured at the equatorial plane using the freehand selection tool, on channels displaying the signal from nuclear (*i.e.*, NLS-2xmCherry) or chromatin-associated proteins (i.e. H3.1- or H3.3-FP fusions). H3.1-eGFP levels were quantified in the nucleus of infected cells (IC) and of directly adjacent, neighboring cells (NC) present on the same z-stack in the same cortical layer. For this, an outline was manually drawn around each nucleus at the equatorial plane and area, integrated density, and mean gray values were measured. Three oval selections were drawn adjacent to all measured nuclei to be used for normalization against background fluorescence. The corrected total nuclear fluorescence (CTNF)=integrated density – (area of selected nucleus ×mean fluorescence of background readings) was calculated according to *McCloy et al., 2014* using Microsoft Excel. The same procedure was applied to quantify nuclear fluorescence associated with the pKNOLLE::D-box-3xVenus-NLS construct. The number of mCitrine-CENH3 signals in individual nuclei and mitotic figures was manually determined across image stacks using the multi-point tool. The length of CENH3 fluorescent doublets was measured between the two most distant pixels using the straight-line tool.

## β-galactosidase enzymatic assay

To evaluate the colonization status of WT roots without or with additional expression of *NF-YA1*, composite plants were generated by hairy root transformation and inoculated with the Sm2011-*lacZ* strain. Transgenic roots accumulating betalain in root tips and lateral roots (based on the pSlEX-T1::RUBY transformation marker; *He et al., 2020*) were selected for further analysis. To reveal rhizobial β-galactosidase activity, transformed root samples were fixed in 1.5% (v/v) glutaraldehyde / Z-buffer (0.1 M sodium phosphate buffer, 10 mM KCl and 1 mM $MgCl_2$, pH 7.4), under vacuum treatment for 15 min and then gently agitated for 45 min at RT. After three washes in Z-buffer, roots were incubated overnight in the dark at 4 °C in Z-buffer containing 5 mM $K_3Fe(CN)_6$, 5 mM $K_4Fe(CN)_6$, and 0.08% (v/v) of 5-bromo-4-chloro-3-indolyl-β-D-galactopyranoside (X-β-Gal; [Carl Roth, Karlsruhe, Germany]). LacZ-stained tissues were washed twice in Z-buffer and stored in 70% EtOH at 4 °C prior to imaging. Whole-root samples were rinsed and mounted on glass slides in deionized water and imaged using the Nikon Eclipse Ni upright microscope (Nikon GmbH, Düsseldorf, Germany), coupled to a DS-Ri2 camera and to the NIS-Elements D (v5.20.00) software. Images were acquired with a 20 x/0.50 or a 40 x/0.75 (Plan Fluor) dry objective.

## Transient expression in *Nicotiana benthamiana* leaves

Four to 5-week-old tobacco (*N. benthamiana*) plants were transiently transformed with *A. rhizogenes* (ARqua1) cells carrying the construct of interest. The first, second, and third fully expanded true leaves were used for infiltration experiments using needle-less syringes. Agrobacteria were resuspended in a MES/KOH infiltration buffer (10 mM, pH 5.6) supplemented with 10 mM $MgCl_2$ and 150 µM acetosyringone. Bacterial cultures used for co-transformations were diluted to a final OD 600 nm of 0.1 for each construct. *Agrobacterium tumefaciens* (GV3101) cells producing the p19 silencing suppressor (*Silhavy et al., 2002*) were systematically included at a final OD 600 nm of 0.05. Upon infiltration, plants were further grown for 64–72 hr before harvesting leaf samples with a biopsy puncher (4 mm diameter) in the distal half of fully infiltrated leaves. Four to five leaf discs were either mounted in sterile tap water and imaged with a confocal microscope, or 10 leaf discs were frozen in liquid nitrogen prior to in vitro transactivation assays.

## Cell division-enabled leaf system (CDELS)

To foster differentiated pavement cells to enter the cell division cycle, non-fluorescently tagged *Arabidopsis* CYCD3;1 was expressed under control of the CaMV 35 S promoter by *Agrobacterium* infiltration (*Xu et al., 2020*). The p35S::CYCD3;1-HA construct was co-transformed with binary vectors allowing native expression of cell cycle regulated reporters (pH4::GUS, pKNOLLE::GUS, pKNOLLE::eGFP-KNOLLE) and constitutive expression of transcription factors (NF-YA1, NF-YA1$^{mutDBD}$) from Medicago. All plasmids used in the experiments involving the CDEL system are listed in *Supplementary file 1*. Transformed leaves

homogenously expressing nuclear transformation markers included on the corresponding binary vectors were selected using an AXIO Imager.M2m ApoTome.2 light microscope (Zeiss). The activity of HA-tagged CYCD3;1 was verified by the presence of at least one mitotic figure with chromosomes labeled by H3.3-mCherry before harvesting.

## Immunoblot analyses

To detect proteins transiently produced in the CDEL samples, 3 discs from 64 hr post-infiltration leaves were frozen in liquid nitrogen and ground to powder using a Tissue Lyser MM300 (Retsch, Haan, Germany). The powder was resuspended in 120 µl of 2 x loading buffer and heated for 8 min at 95 °C. Equal volumes of samples were loaded and separated by 10% SDS-polyacrylamide gel electrophoresis (PAGE) at constant voltage (80 V). Proteins were transferred to a PVDF membrane (0.2 µm pore size) using the Trans-Blot Turbo semi-dry transfer system (7 min mixed MW MIDI program) (Bio-Rad, Feldkirchen, Germany). After blocking in a 2.5% (w/v) milk/TBST solution for 1 hr at RT, transferred membranes were hybridized overnight at 4 °C with the following primary antibodies: anti-GFP (Takara Bio Europe, Saint-Germain-en-Laye, France; Cat# 632381, RRID:AB_2313808; 1/5000); anti-DsRed (Takara Bio Cat# 632496, RRID:AB_10013483; 1/5000); peroxidase-conjugated anti-HA (Roche Diagnostics, Mannheim, Germany; Cat# 12013819001, RRID:AB_390917; 1/2000); anti-FLAG M2 (Sigma-Aldrich, Merck, Darmstadt, Germany; Cat# F3165, RRID:AB_259529) (1/2000). Membranes were washed and incubated 1 hr at RT with peroxidase-conjugated secondary antibodies (anti-mouse, Sigma-Aldrich Cat# A4416, RRID:AB_258167; anti-rabbit, Sigma-Aldrich Cat# A6154, RRID:AB_258284; 1/2000). Immunoblotted proteins were detected using the Clarity Western ECL Substrate (Bio-Rad). Digital images were acquired using an ECL ChemoCam Imager (Intas Science imaging, Göttingen, Germany) with exposures stopped before saturation.

## Transactivation and quantitative fluorometric GUS assays

*N. benthamiana* leaf samples used for GUS fluorometric assays were harvested 72 hr post-infiltration. The same number of plasmids was used for all co-transformation experiments. A construct driving the accumulation of a nuclear FP (pH3.1::NLS:3xVenus) was included in the absence of exogenous transcription factors or CYCD3;1. Ten leaf discs per transformed leaf were frozen in liquid nitrogen inside a Safe Seal Micro tube containing a stainless-steel bead (5 mm diameter). Samples were ground twice for 30 s (28 /s frequency) using a Tissue Lyser MM300 (Retsch), and the powder was resuspended in 400 µl of cold GUS extraction buffer (50 mM sodium phosphate buffer, pH 7.0, 1 mM EDTA, 10 mM DTT, 0.05% [v/v] Triton X-100 and 1 x Protease Inhibitor Cocktail [Roche]). Crude leaf extracts were incubated for 2 hr at 4 °C on a turning wheel and centrifuged at 16,900 × $g$ for 5 min at 4 °C. Ten µl of the supernatant were used to determine protein concentrations by the dye-binding Bradford assay (Bio-Rad). Twenty-five µg of total proteins in a final volume of 100 µl were pipetted into a white 96 well plate (Greiner Bio-One, Frickenhausen, Germany) before the addition of 200 µl of GUS reaction buffer (50 mM sodium phosphate buffer, pH 7.0, 1 mM EDTA, 1% [v/v] Triton X-100 and 0.5 mM 4-Me thylumbelliferyl-β-D-glucuronide [4 MUG] [Carl Roth]). After 10 min of incubation at 37 °C in the dark, the fluorescence emitted by GUS-mediated hydrolysis of 4 MUG into 4-Methylumbelliferone (4-MU) was measured using a microplate reader (POLARstar Omega, BMG LABTECH, Ortenberg, Germany). Fluorescence was recorded at 360 nm excitation and 460 nm emission filter (gain 500, 20 flashes per well) every 5 min over a time period of 130–180 min. Samples were run in technical triplicates, and blank measurements (100 µl extraction buffer +200 µl reaction buffer) were performed to correct for any nonenzymatic hydrolysis of 4 MUG. Corrected GUS activities (4-MU fluorescence units [FU] min$^{-1}$) were calculated as: $A_{GUS} = \Delta F / \Delta t = (F1-F0) / \Delta t$ where $\Delta F$ is the difference in fluorescence intensities measured in the linear range of the fluorescence curves over a 15 min period of time ($\Delta t$). Values were normalized to 1 µg of protein content by using the protein concentration determined via Bradford assay. Two to five independent experiments were performed with at least three leaves from two different plants analyzed.

## Data visualization and statistical analysis

Alignment of multiple protein sequences was performed using Jalview (version 2.11.2.6; *Procter et al., 2021*). Statistical analyses and generation of figures were performed using the GraphPad Prism software (versions 9.5.1 and 10.4.1) (GraphPad Software Inc). Normal distribution of data was tested using

the Shapiro-Wilk and Kolmogorov-Smirnov normality tests. A Brown-Forsythe and Welch ANOVA test followed by Dunnett's multiple T3 comparisons or an unpaired t-test with Welch's correction were applied as parametric tests. Kruskal-Wallis followed by Dunn's multiple comparisons or Mann-Whitney tests were applied as non-parametric tests. Boxes in the box-and-whiskers plots extend from the 25th to 75th percentiles, and whiskers range from the smallest to the largest values. Middle horizontal lines are plotted at the median. Sample size n, statistical tests, and significance levels are provided in the figure legends. Source data and results of the statistical analyses are provided as supplement files associated with the corresponding figure.

### *Medicago truncatula* gene identifiers

Accession numbers for the Medicago genes used in this study are provided from the following genome portals, Phytozome (Mt4.0v1 version; https://phytozome-next.jgi.doe.gov/) and /*Medicago truncatula* A17 r5.0 (A17 r5.1.9 version; https://medicago.toulouse.inra.fr/MtrunA17r5.0-ANR/), respectively: *H3.1* (Medtr8g092720/MtrunA17_Chr8g0383781); *H3.1* (2) (Medtr8g103245/MtrunA17_Chr8g0390361); *H3.3* (Medtr4g097175/MtrunA17_Chr4g0054161); *H4* (Medtr7g099610/MtrunA17_Chr7g0262791); *CDT1a* (Medtr1g070170/MtrunA17_Chr1g0184201); *KNOLLE* (Medtr5g012010/MtrunA17_Chr5g0398891); *CENH3* (Medtr8g027840/MtrunA17_Chr8g0347231); repressor-type *MYB3R* (Medtr7g061330/MtrunA17_Chr7g0238631); *NF-YA1* (Medtr1g056530/MtrunA17_Chr1g0177091); *NPL* (Medtr3g086320/MtrunA17_Chr3g0123331); *ENOD11* (Medtr3g415670/MtrunA17_Chr3g0082994).

## Acknowledgements

We would like to thank the entire team for the constant and fruitful input into the project and especially Eija Schulze and Carmen Schubert for their excellent experimental support. Furthermore, we greatly appreciated the technical help of Zenglin Li (University of Freiburg, now Northwest A&F University, China) and Philipp Schwenk (University of Freiburg, Germany) on GUS fluorometric assays. Jean Keller (Université Toulouse III – Paul Sabatier, France) conducted a comprehensive phylogeny on MYB3R, MEME and FIMO searches on MSA motifs. Katharina Schiessl (University of Cambridge, UK) provided the *nf-ya1-1* seeds and fruitful comments throughout the project. We also thank the staff of the Life Imaging Center (LIC) in the Hilde Mangold House (HMH) of the Albert-Ludwigs-University of Freiburg for the help with their confocal microscopy resources, and the excellent support in image recording. The microscopes are operated by the Microscopy and Image Analysis Platform (MIAP) and the Life Imaging Center (LIC), Freiburg.

## Additional information

### Funding

| Funder | Grant reference number | Author |
| --- | --- | --- |
| Bill and Melinda Gates Foundation | OPP1172165 | Morgane Batzenschlager<br>Beatrice Lace<br>Franck Anicet Ditengou<br>Thomas Ott |
| China Scholarship Council | 20170808001 | Chao Su |
| Gates Agricultural Innovations | G119217 | Morgane Batzenschlager<br>Jule Salfeld<br>Thomas Ott |
| Deutsche Forschungsgemeinschaft | 431626755 | Chao Su<br>Thomas Ott |
| Deutsche Forschungsgemeinschaft | 403222702 | Beatrice Lace<br>Thomas Ott |
| Deutsche Forschungsgemeinschaft | 39093984 | Ning Zhang<br>Thomas Laux<br>Thomas Ott |

| Funder | Grant reference number | Author |
|---|---|---|
| Deutsche Forschungsgemeinschaft | 414136422 | Thomas Ott |

The funders had no role in study design, data collection and interpretation, or the decision to submit the work for publication.

## Author contributions

Morgane Batzenschlager, Conceptualization, Data curation, Formal analysis, Supervision, Investigation, Visualization, Writing – original draft, Project administration, Writing – review and editing; Beatrice Lace, Supervision, Investigation, Writing – review and editing; Ning Zhang, Chao Su, Sabrina Egli, Pascal Krohn, Jule Salfeld, Franck Anicet Ditengou, Investigation, Writing – review and editing; Anna Boiger, Investigation; Thomas Laux, Supervision, Funding acquisition, Project administration, Writing – review and editing; Thomas Ott, Conceptualization, Supervision, Funding acquisition, Writing – original draft, Project administration, Writing – review and editing

## Author ORCIDs

Morgane Batzenschlager https://orcid.org/0000-0002-7559-9172
Beatrice Lace https://orcid.org/0000-0002-4732-573X
Chao Su https://orcid.org/0000-0003-4084-0808
Thomas Ott https://orcid.org/0000-0002-4494-9811

Reviewer #1 (Public Review): https://doi.org/10.7554/eLife.88588.3.sa1
Reviewer #2 (Public Review): https://doi.org/10.7554/eLife.88588.3.sa2
Author response https://doi.org/10.7554/eLife.88588.3.sa3

# Additional files

## Supplementary files

MDAR checklist

Supplementary file 1. Table S1:Binary plasmids generated using Golden Gate cloning.

## Data availability

All data generated and analyzed during this study are included in the manuscript and supporting files. All sequences have been obtained from publicly available repositories and indicated in the Materials and Methods section. All source data linked to the corresponding figures contain the numerical data used to generate the figures. All material generated within this study can be obtained from the corresponding authors upon request. Plasmids obtained from other laboratories as indicated in the respective sections need to be requested from the original producer.

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
