## [Editor Report · eLife Assessment]

This is a **fundamental** cell biological study of host responses during symbiotic microbial infection of plants. **Compelling** imaging-based approaches using genetically encoded cell cycle markers show that in Medicago truncatula root cortex cells, early rhizobial infection events are associated with cell-cycle re-entry, but once the infection is established, host cells exit the cell cycle. The work will be of interest to a wide range of readers working in fields from development and cell biology to plant-microbe interactions.

---

## [Referee Report · Reviewer #1 (Public Review)]

Many studies reported findings implying that rhizobial infection is associated with cell cycle re-entry and progression, however, our understanding has been fragmented. This study provides exciting new insights as it represents a comprehensive description of the cell cycle progression during early stages of nodulation using fluorescence markers.

To briefly summarize, the authors first monitor H3.1 / H3.3 replacement to distinguish between replicating (S phase) and non-replicating cells to show that M. truncatula cortex cells along the bacterial infection thread are non-replicating (while neighbors enter the S phase). Nuclear size measurements revealed that these non-replicative cells are in the post-replicative stage (G2) rather than in the pre-replicative G1 phase, which the authors confirm with the Plant Cell Cycle Indicator (PlaCCI) fluorescent marker to track cell cycle progression in more detail. Cortex cells in the trajectory of the infection thread did not accumulate the late G2 marker of the PlaCCI nor the G2/M marker KNOLLE, indicating that these cells indeed remain in G2. Because nuclear size measurements indicated that infected cells are polyploid, the authors used the centromere histone marker CENH3 to determine chromosome number. They find that cortex cells giving rise to the nodule primordium are endomitotic and tetraploid, probably because their cell cycle is halted at centromere separation. Although not a focus of this manuscript, the authors also use their fluorescent tools to track cell cycle progression during arbuscular mycorrhiza symbiosis. They confirm that infected cells transition from a replicating to a non-replicating state (H3.1 to H3.3) with progressing development of the arbuscules. In addition, the CENH3 marker confirms previous findings that cortex cells infected by fungi are endocycling (i.e., DNA synthesis without segregation of replicated parts). This represents an important confirmation of previous findings and contrasts with the situation during nodulation symbiosis, where chromosomes separate after replication.

In general, all microscopy images are of very high quality and support the authors' conclusions. While individually each set of fluorescent markers has its limitations, combined they constitute a powerful tool to track various stages of cell cycle progression in individual root cells during symbiosis. Overall, this is a very strong manuscript that comprehensively elucidates root cell cycle changes during microbial infection.

---

## [Referee Report · Reviewer #2 (Public Review)]

Cell cycle control during nitrogen-fixing symbiosis is an important topic, but our understanding of the process is poor and lacks resolution, as the nodule is a complex organ with many cell types that undergo profound changes. The authors aim to define the cell cycle state of individual plant cells in the emerging nodule primordium, as a transcellular infection thread passes through the meristem to reach cells deep in the incipient nodule and releases bacteria to form symbiosomes. The authors used a number of cell cycle reporters, such as different Histone 3 variants and cyclins, to follow cell cycle progress in exquisite detail. They showed that the host cells in the path of an infection thread exhibit a cell fate distinct from their immediate neighbors: after entering the S phase similar to their neighbors, these cells exit the cell cycle and enter a special differentiated state. This is likely an important shift that allows the proper passage of the infection thread. Although definitive proof needs more investigation, they showed that a pioneering transcription factor, NF-YA1, likely represses these endoreduplicated cells from completing the cell cycle.

---

## [Author Response]

The following is the authors’ response to the original reviews

**Reviewer #1 (Public Review):**
(…) In my view, the part about NF-YA1 is less strong - although I realize this is a compelling candidate to be a regulator of cell cycle progression, the experimental approaches used to address this question falls a bit short, in particular, compared to the very detailed approaches shown in the rest of the manuscript. The authors show that the transcription factor NF-YA1 regulates cell division in tobacco leaves; however, there is no experimental validation in the experimental system (nodules). All conclusions are based on a heterologous cell division system in tobacco leaves. The authors state that NF-YA1 has a nodule-specific role as a regulator of cell differentiation. I am concerned the tobacco system may not allow for adequate testing of this hypothesis.

Reviewer #1 makes a valid point by asking to focus the manuscript more explicitly on the role of NF-YA1 as a differentiation factor in a symbiotic context. We have now addressed this formally and experimentally.

The involvement of A-type NF-Y subunits in the transition to the early differentiation of nodule cells has been documented in model legumes through several publications that we refer to in the revised version of the discussion (lines 617/623). We fully agree that the CDEL system, because it is heterologous, does not allow us more than to propose a parallel explanation for these observations - i.e_._, that the Medicago NF-YA1 subunit presumably acts in post-replicative cell-cycle regulation at the G2/M transition. Considering your recommendations and those of reviewer #2, we sought to support this conclusion by testing the impact of localized over-expression of *NF-YA1* on cortical cell division and infection competence at an early stage of root colonization. The results of these experiments are now presented in the new Figure 9 and Figure 9-figure supplement 1-5 and described from line 435 to 495.

With the fluorescent tools the authors have at hand (in particular tools to detect G2/M transition, which the authors suggest is regulated by NF-YA1), it would be interesting to test what happens to cell division if NF-YA1 is over-expressed in Medicago roots?

To limit pleiotropic effects of an ectopic over-expression, we used the symbiosis-induced, *ENOD11* promoter to increase *NF-YA1* expression levels more specifically along the trajectory of infected cells. We chose to remain in continuity with the experiments performed in the CDEL system by opting for a destabilized version of the *KNOLLE* transcriptional reporter to detect the G2/M transition. The results obtained are presented in Figure 9B (quantification of split infected cells), in Figure 9-figure supplement 1B (*ENOD11* expression profile), in Figure 9-figure supplement 3B (representative confocal images) and Figure 9-figure supplement 4D (quantification of pKNOLLE reporter signal). There, we show that mitosis remains inhibited in cells accommodating infection threads, but is completed in a higher proportion of outer cortical cells positioned on the infection trajectory, where *ENOD11* gene transcription is active before their physical colonization.

Based on NF-YA1 expression data published previously and their results in tobacco epidermal cells, the authors hypothesize that NF-YA regulates the mitotic entry of nodule primordial cells. Given that much of the manuscript deals with earlier stages of the infection, I wonder if NF-YA1 could also have a role in regulating mitotic entry in cells adjacent to the infection thread?

The expression profile of *NF-YA1* at early stages of cortical infection (Laporte et al., 2014) is indeed similar to the one of *ENOD11* (as shown in Figure 9-figure supplement 1C) in wild-type Medicago roots, with corresponding transcriptional reporters being both activated in cells adjacent to the infection thread. Under our experimental conditions, additional expression of *NF-YA1* (driven by the *ENOD11* promoter) in these neighbouring cells did not impact their propensity to enter mitosis and to complete cell division. These results are presented in Figure 9-figure supplement 4D (quantification of pKNOLLE reporter signal) and Figure 9-figure supplement 5 (quantification of split neighbouring cells).

**Reviewer #1 (Recommendations For The Authors):**
- In the first part, images show the qualitative presence/absence of H3.1 or H3.3 histones.Upon closer inspection, many cells seem to have both histones. In Fig1-S1 for example (root meristem), it is evident that there are many cells with low but clearly present H3.1 content in the green channel; however, in the overlay, the green is lost and H3.3 (pink) is mainly visible. What does this mean in terms of the cell cycle?

We fully agree with reviewer #1 on these points. Independent of whether they have low or high proliferation potential, most cells retain histone H3.1 particularly in silent regions of the genome, while H3.3 is constitutively produced and enriched at transcriptionally active regions. When channels are overlaid, cells in an active proliferation or endoreduplication state (in G1, S or G2, depending on the size of their nuclei) will appear mainly "green" (H3.1-eGFP positive). Cells with a low proliferation potential (e.g., in the QC), G2-arrested (e.g., IT-traversed) or terminally differentiating (e.g., containing symbiosomes or arbuscules) will appear mainly "magenta" (H3.1-low, medium to high H3.3-mCherry content).

Furthermore, all nodule images only display the overlay image, and individual fluorescence channels are not shown. Does the same masking effect happen here? It may be helpful to quantify fluoresce intensity not only in green but also in red channels as done for other experiments.

Quantifying fluorescence intensity in the mCherry channel may indeed help to highlight the likely replacement of H3.1-eGFP by H3.3-mCherry in infected cells, as described by Otero and colleagues (2016) at the onset of cellular differentiation. However, the quantification method as established (i.e., measuring the corrected total nuclear fluorescence at the equatorial plane) cannot be applied, most of the time, to infected cells' nuclei due to the overlapping presence of mCherry-producing *S. meliloti* in the same channel (e.g., in Figure 2B). Nevertheless, and to avoid this masking effect when the eGFP and mCherry channels are overlaid, we now present them as isolated channels in revised Figures 1-3 and associated figure supplements. As the cell-wall staining is regularly included and displayed in grayscale, we assigned to both of them the Green Fire Blue lookup table, which maps intensity values to a multiple-colour sequential scheme (with blue or yellow indicating low or high fluorescence levels, respectively). We hope that this will allow a better appreciation of the respective levels of H3.1- and H3.3-fusions in our confocal images.

- Fig 1 B - it is hard to differentiate between S. meliloti-mCherry and H3.3-mCherry. Is there a way to label the different structures?

In the revised version of Figure 1B, we used filled or empty arrowheads to point to histone H3-containing nuclei. To label rhizobia-associated structures, we used dashed lines to delineate nodule cells hosting symbiosomes and included the annotation “IT” for infection threads. We also indicated proliferating, endoreduplicating and differentiating tissues and cells using the following annotations: “CD” for cell division, “En” for endoreduplication and “TD” for terminal differentiation. All annotations are explained in the figure legend.

- Fig 1 - supplement E and F - no statistics are shown.

We performed non-parametric tests using the latest version of the GraphPad Prism software (version 10.4.1). Stars (Figure 1-figure supplement 1F) or different letters (Figure 1-figure supplement 1G) now indicate statistically significant differences. Results of the normality and non-parametric tests were included in the corresponding Source Data Files (Figure 1 – figure supplement 1 – source data 1 and 2). We have also updated the compact display of letters in other figures as indicated by the new software version. The raw data and the results of the statistical analyses remain unchanged and can be viewed in the corresponding source files.

- Fig 2 A - overview and close-up image do not seem to be in the same focal plane. This is confusing because the nuclei position is different (so is the infection thread position).

We fully agree that our former Figure may have confused reviewers #1 and #2 as well as readers. Figure 2A was designed to highlight, from the same nodule primordium, actively dividing cells of the inner cortex (optical section z 6-14) and cells of the outer cortex traversed, penetrated by or neighbouring an infection thread (optical section z 11-19). We initially wanted to show different magnification views of the same confocal image (i.e_._, a full-view of the inner cortex and a zoomed-view of the outer layers) to ensure that audiences can identify these details. In the revised version of Figure 2A, we displayed these full- and zoomed-views in upper and lower panels, respectively and we removed the solid-line inset to avoid confusion.

- Fig 1A and Fig 2E could be combined and shown at the beginning of the manuscript. Also, consider making the cell size increase more extreme, as it is important to differentiate G2 cells after H3.1 eviction and cells in G1. You have to look very closely at the graph to see the size differences.

We have taken each of your suggestions into account. A combined version of our schematic representation with more pronounced nuclei size differences is now presented in Figure 1A.

- Fig. 3 C is difficult to interpret. Can this be split into different panels?

We realized that our previous choice of representation may have been confusing. Each value corresponds only to the H3.1-eGFP content, measured in an infected cell and reported to that of the neighbouring cell (IC / NC) within individual root samples. Therefore, we removed the green-magenta colour code and changed the legend accordingly. We hope that these slight modifications will facilitate the interpretation of the results - namely, that the relative level of H3.1 increases significantly in infected cells in the selected mutants compared to the wild-type. This mode of representation also highlights that in the mutants, there are more individual cases where the H3.1 content in an infected cell exceeds that of the neighbouring cell by more than two times. These cases would be masked if the couples of infected cells and associated neighbours would be split into different panels as in Figure 3B.

- Line 357/359. I assume you mean ...'through the G2 phase can commit to nuclear division'.

We have edited this sentence according to your suggestion, which now appears in line 370.

**Reviewer #2 (Recommendations For The Authors):**
Cell cycle control during the nitrogen-fixing symbiosis is an important question but only poorly understood. This manuscript uses largely cell biological methods, which are always of the highest quality - to investigate host cell cycle progression during the early stages of nodule formation, where cortical infection threads penetrate the nodule primordium. The experiments were carefully conducted, the observations were detail oriented, and the results were thought-provoking. The study should be supported by mechanistic insights.(1) One thought provoked by the authors' work is that while the study was carried out at an unprecedented resolution, the relationship between control of the cell cycle and infection thread penetration remains correlative. Is this reduced replicative potential among cells in the infection thread trajectory a consequence of hosting an infection thread, or a prerequisite to do so?

We understand and share the point of view of reviewer #2. At this stage, we believe that our data won’t enable us to fully answer the question, thus this relationship remains rather correlative. The reasons are that (1) the access to the status of cortical cells below C2 is restricted to fixed material and therefore only represents a snapshot of the situation, and (2) we are currently unable to significantly interfere with mechanisms as intertwined as cell cycle control and infection control. What we can reasonably suggest from our images is that the most favorable window of the cell cycle for cells about to be crossed by an infection thread is post-replicative, i.e., the G2 phase. Typical markers of the G2 phase were recurrently observed at the onset of physical colonization – enlarged nucleus, containing less histone H3.1 than neighbouring cells in S phase (e.g., in Figure 2A). Reaching the G2 phase could therefore be a prerequisite for infection (and associated cellular rearrangements), while prolonged arrest in this same phase is likely a consequence of transcellular passage towards a forming nodule primordium.

More importantly, in either scenario, what is the functional significance of exiting the cell cycle or endocycle? By stating that "local control of mitotic activity could be especially important for rhizobia to timely cross the middle cortex, where sustained cellular proliferation gives rise to the nodule meristem" (Line 239), the authors seem to believe that cortical cells need to stop the cell cycle to prepare for rhizobia infection. This is certainly reasonable, but the current study provides no proof, yet. To test the functional importance of cell cycle exit, one would interfere with G2/M transition in nodule cells, and examine the effect on infection.

We fully agree with reviewer #2 that the functional importance of a cell-cycle arrest on the infection thread trajectory remains to be demonstrated. Interfering with cell-cycle progression in a system as complex and fine-tuned as infected legume roots certainly requires the right timing – at the level of the tissue and of individual cells; the right dose; and the right molecular player(s) (i.e., bona fide activators or repressors of the G2/M transition). Using the symbiosis-specific *NPL* promoter, activated in the direct vicinity of cortical infection threads (Figure 9-figure supplement 1B), we tried to force infectable cells to recruit the cell division program by ectopically over-expressing the Arabidopsis *CYCD3.1*, “mimicking” the CDEL system. So far, this strategy has not resulted in a significant increase in the number of uninfected nodules in transgenic hairy roots - though the effect on symbiosome release remains to be investigated. Provided that a suitable promoter-cell cycle regulator combination is identified, we hope to be able to answer this question in the future.

Given that the authors have already identified a candidate, and showed it represses cell division in the CDEL system, not testing the same gene in a more relevant context seems a lost opportunity. If one ectopically expressed NY-YA1 in hairy roots, thus repressing mitosis in general, would more cells become competent to host infection threads? This seems a straightforward experiment and readily feasible with the constructs that the authors already have. If this view is too naive, the authors should explain why such a functional investigation does not belong in this manuscript.

Reviewer #2's point is entirely valid, and we decided to address it through additional experiments. To avoid possible side effects on development by affecting cell division in general, we placed *NF-YA1* under control of the symbiosis-induced *ENOD11* promoter. Based on the results obtained in the CDEL system, the pENOD11::FLAG-NF-YA1 cassette was coupled to a destabilized version of the *KNOLLE* transcriptional reporter to detect the G2/M transition. Competence for transcellular infection was maintained upon local *NFYA1* overexpression, the latter leading to a slight (non-significant) increase in the number of infected cells per cortical layer. These results are presented in Figure 9-figure supplement 3A-B (representative confocal images) and in Figure 9-figure supplement 4A-

G.

(1b) A related comment: on Line 183, it was stated that "The H3.1-eGFP fusion protein was also visible in cells penetrated but not fully passed by an infection thread". Presumably, the authors were talking about the cell marked by the arrowhead. But its H3.1-GFP signal looks no different from the cell immediately to its left. It is hard to say which cells are ones "preparing for intracellular infection pass through S-phase", and which ones are just "regularly dividing cortical cells forming the nodule primordium". What can be concluded is that once a cell has been fully transversed by an infection thread, its H3.1 level is low. Whether this is the cause or consequence of infection cannot be resolved simply by timing the appearance or disappearance of H3.1-GFP.

We basically agree with comment 1b. In an unsynchronized system such as infected hairy roots, it is challenging to detect the event where a cell is penetrated, but not yet completely crossed by an infection thread. What we wanted to emphasize in Figure 2A, is that host cells in the path of an infection thread re-enter the cell cycle and pass through S-phase just as their neighbours do (as pointed out by reviewer #2 in his summary). The larger nucleus with slightly lower H3.1-eGFP signal than the neighbouring cell (as indicated by the use of the Green Fire Blue lookup table) suggests that the infected cell marked by the arrowhead in Figure 2A is actually in the G2 phase. The main difference is indeed that cells allowing complete infection thread passage exit the cell cycle and largely evict H3.1 while their neighbours proceed to cell division (as exemplified by PlaCCI reporters in Figure 4CD and the new Figure 5-figure supplement 2). Whether cell-cycle exit in G2 is a cause, or a consequence of cortical infection is a question that cannot be easily answered from fixed samples, which is a limitation of our study.

(2) The authors have convincingly demonstrated that cortical cells accommodating infection threads exit the cell cycle, inhibit cell division, and down-regulate KNOLLE expression. How do these observations reconcile with the feature called the pre-infection thread? The authors devoted one paragraph to this question in the Discussion, but this does seem sufficient given that the pre-infection thread is a prominent concept. Is the resemblance to the cell division plane superficial, or does it reflect a co-option of the normal cytokinesis machinery for accommodating rhizobia?

From our point of view, cortical cells forming pre-infection threads are likely in an intermediate state. PIT structures undoubtedly share many similarities with cells establishing a cell division plane. The recruitment of at least some of the players normally associated with cytokinesis has been demonstrated and is consistent with the maintenance of infectable cells in a pre-mitotic phase in Medicago, as discussed in lines 558 to 568. We nevertheless think that the arrest of the cell cycle in the G2 phase, presumably occurring in crossed cortical cells, constitutes an event of cellular differentiation and specialization in transcellular infection.

The following are mainly points of presentation and description:(3) Line 158: I can't see "subnuclear foci" in Figure 1-figure supplement 1C-E. However, they are visible in Fig. 1C.

We hope that presenting the eGFP and mCherry channels in separate panels and assigning them the Green Fire Blue colour scheme provides better visibility and contrast of these detailed structures. We now refer to Figure 1C in addition to Figure 1–figure supplement 1E in the main text (line 161).

(4) Line 160: The authors should outline a larger region containing multiple QC cells, rather than pointing to a single cell, as there are other areas in the image containing cells with the same pattern.

We updated Figure 1-figure supplement 1E accordingly.

(5) Fig. 1B should include single channels, since within a single plant cell, the nucleus, the infection thread, and sometimes symbiosomes all have the same color. This makes it hard to see whether the nuclei in these cells are less green, or are simply overwhelmed by the magenta color.

To improve the readability of Figure 1B and to address suggestions from individual reviewers, we now include separate channels and have annotated the different structures labeled by mCherry.

(6) Fig. 2A: the close-up does not match the boxed area in the left panel. Based on the labeling, it seems that the two panels are different optical sections. But why choose a different optical depth for the left panel? This can be disorienting to the author, because one expects the close-up to be the same image, just under higher magnification.

We fully agree that our previous choice of representation may have been confusing. As we also specified to reviewer #1, we wanted to show a full-view of proliferating cells in the inner cortex and a zoomed-view of infected cells in the outer layers of the same nodule primordium. In the revised version of Figure 2A, we displayed these full- and zoomedviews in separate panels and removed the boxed area to avoid confusion.

(7) Figure 2-figure supplement 1B: the cell indicated by the empty arrowhead has a striking pattern of H3.1 and H3.3 distribution on condensed chromosomes. Can you comment on that?

Reviewer #2 may be referring to the apparent enrichment of H3.3 at telomeres, previously described in Arabidopsis, while pericentromeric regions are enriched in H3.1. This distribution is indeed visible on most of the condensed chromosomes shown in Figure 2-figure supplement 1B. We included this comment in the corresponding caption.

(8) Fig. 4: It is not very easy to distinguish M phase. Can the authors describe how each phase is supposed to look like with the reporters?

We agree with reviewer #2 and attempted to improve Figure 4, which is now dedicated to the Arabidopsis PlaCCI reporter. ECFP, mCherry, and YFP channels were presented separately and the corresponding cell-cycle phases (in interphase and mitosis) were annotated. The Green Fire Blue lookup table was assigned to each reporter to provide the best visibility of, for example, chromosomes in early prophase. We included a schematic representation corresponding to the distribution of each reporter, using the colors of the overlaid image to facilitate its interpretation.

(9) Line 298: what is endopolyploid? This term is used at least three times throughout the manuscript. How is it different from polyploid?

In the manuscript, we aimed to differentiate the (poly)ploidy of an organism (reflecting the number of copies of the basic genome and inherited through the germline) from endopolyploidy produced by individual somatic cells. As reviewed by Scholes and Paige, polyploidy and endopolyploidy differ in important ways, including allelic diversity and chromosome structural differences. In the *Medicago truncatula* root cortex for example, a tetraploid cell generated via endoreduplication from the diploid state would contain at most two alleles at any locus. The effects of endopolyploidy on cell size, gene expression, cell metabolism and the duration of the mitotic cell cycle are not shared among individual cells or organs, contrasting to a polyploid individual (Scholes and Paige, 2015).

See Scholes, D. R., & Paige, K. N. (2015). Plasticity in ploidy : A generalized response to stress. Trends in Plant Science, 20(3), 165‑175. https://doi.org/10.1016/j.tplants.2014.11.007

(10) Line 332: "chromosomes on mitotic figures" - what does this mean?

Reviewer #2 is right to point out this redundant wording. Mitotic “figures” are recognized, by definition, based on chromosome condensation. We now use the term "mitotic chromosomes" (line 344).

(11) Fig. 6A: could the authors consider labeling the doublets, at least some of them? I understand that this nucleus contains many doublets. However, this is the first image where one is supposed to recognize these doublets, and pointing out these features can facilitate understanding. Otherwise, a reader might think the image is comparable to nuclei with no doublets in the rest of the figure.

Following this suggestion, five of these doublets are now labeled in Figure 7A (formerly Figure 6A).